# Data-Agnostic Augmentations for Unknown Variations: Out-of-Distribution Generalisation in MRI Segmentation

Puru Vaish[*1]                                              P.VAISH@UTWENTE.NL
Felix Meister[2]                        FELIX.MEISTER@SIEMENS-HEALTHINEERS.COM
Tobias Heimann[2]                     TOBIAS.HEIMANN@SIEMENS-HEALTHINEERS.COM
Christoph Brune[1]                                         C.BRUNE@UTWENTE.NL
Jelmer M. Wolterink[1]                            J.M.WOLTERINK@UTWENTE.NL

[1] *Department of Applied Mathematics, Technical Medical Centre, University of Twente*

[2] *Digital Technology and Innovation, Siemens Healthineers, Erlangen, Germany*

**Editors:** Accepted for publication at MIDL 2025

## Abstract

Medical image segmentation models are often trained on curated datasets, leading to performance degradation when deployed in real-world clinical settings due to mismatches between training and test distributions. While data augmentation techniques are widely used to address these challenges, traditional visually consistent augmentation strategies lack the robustness needed for diverse real-world scenarios. In this work, we systematically evaluate alternative augmentation strategies, focusing on MixUp and Auxiliary Fourier Augmentation. These methods mitigate the effects of multiple variations without explicitly targeting specific sources of distribution shifts. We demonstrate how these techniques significantly improve out-of-distribution generalization and robustness to imaging variations across a wide range of transformations in cardiac cine MRI and prostate MRI segmentation. We quantitatively find that these augmentation methods enhance learned feature representations by promoting separability and compactness. Additionally, we highlight how their integration into nnU-Net training pipelines provides an easy-to-implement, effective solution for enhancing the reliability of medical segmentation models in real-world applications.

**Keywords:** MRI, segmentation, data augmentation, generalisation, robustness

## 1. Introduction

Medical image analysis requires deep learning models that are accurate, robust, and generalize well to new and unseen data. However, when deployed in real-world scenarios, deep neural networks often suffer performance degradation (Hendrycks and Dietterich, 2019; Kamann and Rother, 2021). This generalization gap can be attributed to a range of factors, including variations in patient populations, differences in image acquisition, and imaging artefacts. Among strategies to improve generalization are data augmentation techniques like intensity shifts, affine transforms, and noise addition (Garcea et al., 2023; Goceri, 2023). As they can demonstrably improve out-of-distribution generalization (Boone et al., 2023), they are among the standard set of augmentations used in many deep learning segmentation models, such as nnU-Net (Isensee et al., 2021).

However, standard augmentation strategies cannot cover more complex underlying image formation mechanisms, which in MRI could include bias fields due to coil miscalibration,

---

* Corresponding author

Rician noise when MRI is taken at higher resolutions, ghosting artefacts, or random RF spikes during acquisition (see Fig. 1). Artifact-specific augmentation policies (Boone et al., 2023) or pre-processing methods such as bias field correction might mitigate this problem, but such processes are not guaranteed to exist, for instance, Rician noise, ghosting and unseen issues due to data mishandling (Shimron et al., 2022), or be missed entirely in automated workflows. Furthermore, explicitly anticipating all possible variations is often infeasible. Hence, as an alternative, augmentation strategies that are not specifically designed for any variation and yet manages to mitigate the effect of multiple variations would greatly benefit medical imaging with deep learning.

In this work, we systematically investigate general, *data-agnostic* augmentation strategies, namely MixUp (Zhang et al., 2018) and Auxiliary Fourier Augmentation (AFA) (Vaish et al., 2024). By data-agnostic, we mean augmentations that do not seek to maintain the visual consistency of the data being augmented. We demonstrate the effect of these techniques in nnU-Net models for segmentation of cardiac cine MRI and prostate MRI. Neither MixUp nor AFA explicitly addresses specific sources of variation in these data, yet we show how they improve segmentation performance in various out-of-distribution generalization settings. Moreover, we include an analysis of the learned feature representations, showing improved structure and interoperability when MixUp and AFA are used. Our findings provide new insights into the effectiveness and limitations of these augmentation methods in medical image analysis scenarios and show that MixUp and AFA can improve the performance of deep neural networks in multiple tasks and generalization settings.

## 2. Materials and Methods

### 2.1. Data

We investigate to what extent data augmentation strategies can mitigate the effects of distribution shifts in MRI. For each of cardiac cine MRI and bi-parametric prostate MRI we perform two tests. First, to test the effect of real-world distribution shifts, we include two separate datasets, allowing us to consider differences within and between datasets. Second, as it is hard to quantify a generalisation gap between real world datasets due to various MR image characteristics, we modify the test set using controlled MRI transformations allowing us to isolate failures to specific MR image variations.

**Cardiac cine MR** We use the Automated Cardiac Diagnosis Challenge (ACDC) (Bernard et al., 2018), which contains 150 cardiac cine MRI scans (100 training, 50 test) acquired at Hospital of Dijon, France (in-plane resolution 1.37-1.67mm, slice thickness 5-10mm). As an external test set used to measure generalization performance, we include the M&Ms (Campello et al., 2021) test set of 268 scans with different pathologies from multiple centers (in-plane resolution 0.85-1.45mm, slice thickness 10mm). In both ACDC and M&Ms, manual annotations of the left ventricle (LV), myocardium (MYO), and right ventricle (RV) are provided in end-diastolic (ED) and end-systolic (ES) frames.

**Prostate MRI** We include prostate bi-parametric MRI (bpMRI) scans from the Prostate 158 (P158) dataset (Adams et al., 2022), which has 139 scans for training and 19 scans for testing. The DWI images were acquired at b-values ranging from 50 to 1000 $s/mm^3$ and a high b-value of 1400 $s/mm^3$. In addition, as an external test set to measure generalization,

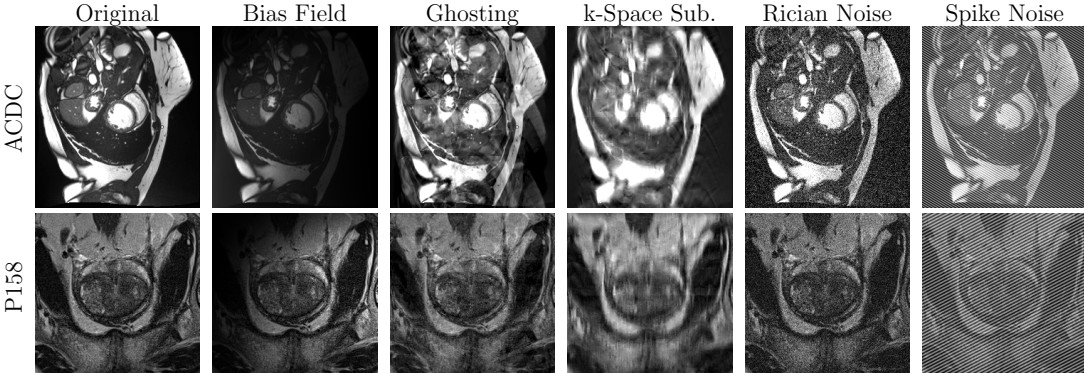

Figure 1: Corruptions (severity level 3) in cardiac cine MRI images and the T2w channel of prostate bpMRI images as a result of our image variation model.

we use the ProstateX (PX) dataset (Armato et al., 2018), which includes 141 test scans where DWI scans were acquired with 3 b-values (50, 400, and 800 $s/mm^2$), and a computed apparent diffusion coefficient map. Both datasets provide T2w scans at an in-plane resolution of 1.45-1.5 mm, ADC maps at 0.45-0.5 mm, and slice thicknesses of 3-4mm. Segmentation masks of the transitional zone (TZ) and the prostate peripheral zone (PZ) are available in all images. The test masks for PX are taken from (Xu et al., 2023). Both T2w and ADC are used for model training and evaluation.

## 2.2. Modelling Image Variation

We define five distinct severity levels for each transformation. Severity levels capture various magnitudes of distribution shifts, from mild to severe. As there is a lack of quantitative studies investigating the range of severity of variations in medical imaging data, we followed the same approach as ROOD-MRI (Boone et al., 2023) to determine the generation parameters based on input from experienced MRI technicians. The parameters used to define the severity levels are detailed in our program code and may need to be adapted for other datasets. We apply elastic deformation, isotropic downsampling, anisotropic downsampling, bias field amplification, contrast compression, contrast expansion, ghosting, random motion, Rician noise addition, smoothing, rotation, scaling, spike noise artifacts (radio frequency noise) and k-space subsampling, again at five different levels (1: mild, 5: severe). In total, we apply 14 transformations to each image (see Fig. 1 for examples). It is important to note that these transformations are only applied to the test set and are never seen during the training process. This allows us to systematically study the fragility of these models to unseen MR image variations. This is applied to the test sets of ACDC and P158.

## 2.3. Augmentation Strategies

We conduct experiments in which we train models with three different kinds of augmentation strategies within the nnU-Net framework (Isensee et al., 2021). In all experiments, we keep the pre-processing and post-processing fixed to the default nnU-Net options.

**Base augmentations** By default, nnU-Net employs eight augmentation strategies, namely rotation, scaling, Gaussian noise injection, Gaussian blurring, brightness and contrast adjustments, simulation of low-resolution imaging, gamma correction, and mirroring.

**MixUp** In this strategy, new samples are generated through linear interpolation of pairs of training samples. Formally, given two samples $(x_i, y_i)$, $(x_j, y_j)$ and $\lambda \in [0, 1]$ drawn from a Beta distribution, MixUp creates a new synthetic sample as:

$$x_{\mathrm{mix}} = \lambda x_i + (1 - \lambda)x_j, \quad y_{\mathrm{mix}} = \lambda y_i + (1 - \lambda)y_j.$$

In the original MixUp formulation for classification tasks, images and one-hot encoded labels are both linearly interpolated. Here, we use the exact same strategy, but instead of linearly interpolating between two labels, we interpolate between two one-hot encoded segmentation masks. The loss is then computed using these probability masks as ground truth.

**Auxiliary Fourier Augmentation** (AFA) augments images in the frequency domain under the hypothesis that visual augmentation techniques are unable to cover the vulnerability of neural networks to perturbations in the frequency domain (Vaish et al., 2024). AFA samples frequency basis functions and adds them to the training samples, leaving the label unchanged. Formally, let $\mathcal{F}$ denote the Real Fourier transform operator. For a training sample $(x_i, y_i)$, the $n$-dimensional Fourier transform of $x_i$ is given by $X_i = \mathcal{F}(x_i)$. For a fixed mean, $\mu$, the Fourier spectrum is perturbed by $\alpha \sim \mathrm{Exp}(\mu)$, a real value, at a randomly chosen frequency coordinate $(k_1, k_2, \ldots)$ in the Fourier domain, modifying $X_i$ as:

$$X_i^{\mathrm{aug}}(k_1, k_2, \ldots) = X_i(k_1, k_2, \ldots) + \alpha.$$

The augmented image in the spatial domain, $x_i^{\mathrm{aug}}$, is then obtained by applying the inverse Fourier transform: $x_i^{\mathrm{aug}} = \mathcal{F}^{-1}(X_i^{\mathrm{aug}})$. The model training involves a joint optimization of an AFA-augmented image and a non-AFA-augmented image.

### 2.4. Quantitative Evaluation

We segment all structures separately, namely LV, MYO, RV in cardiac cine MRI, and TZ and PZ in prostate MRI. Results are reported as average Dice Similarity Coefficients (DSC) and $95^{\mathrm{th}}$ percentile Hausdorff Distances (HD95) (as implemented in MONAI (Consortium, 2024)), over all structures, and frames (ED, ES) in cine MRI. For all settings, we perform a 5-fold cross-validation. All predictions are made using an ensemble of five models, which is the default and recommended method to use nnU-Net. To test for statistical significance, we use a paired t-test at $p < 0.05$, on the individual metrics calculated for each sample during testing before ensemble averaging. Structure-wise results are shown in Appendix H.

## 3. Experiments and Results

All hyperparamters, program code and implementation details can be found in Appendix C.

### 3.1. Synthetically Corrupted Images

We train nnU-Net models with different combinations of base augmentations, MixUp, and AFA, for both the ACDC cardiac cine MR dataset and the P158 prostate dataset. For each

Table 1: DSC and HD95 on the original and transformed test set of ACDC and P158 using either using no augmentations or a combination of base, MixUp, and AFA augmentations. Blue and red indicates statistically significant ($p < 0.05$) improved / reduced metrics between models without and with MixUp or AFA using paired $t$-test. **Bold / Bold**-faced numbers indicate the best result for each column.

| | Augmentation | | ACDC | | | | P158 | | | |
| | | | Original | | Transformed | | Original | | Transformed | |
| Base | MixUp | AFA | DSC | HD95 (mm) | DSC | HD95 (mm) | DSC | HD95 (mm) | DSC | HD95 (mm) |
|---|---|---|---|---|---|---|---|---|---|---|
| | | | 89.1 $\sigma$ 5.8 | 5.81 $\sigma$ 5.1 | 75.5 $\sigma$ 24 | 12.7 $\sigma$ 19 | 78.9 $\sigma$ 11 | 4.63 $\sigma$ 2.2 | 70.5 $\sigma$ 20 | 8.23 $\sigma$ 7.8 |
| | ✓ | | 88.9 $\sigma$ 6.3 | 6.08 $\sigma$ 4.9 | 76.0 $\sigma$ 22 | 12.8 $\sigma$ 16 | 78.6 $\sigma$ 15 | 4.99 $\sigma$ 2.3 | 69.6 $\sigma$ 22 | 8.29 $\sigma$ 8.9 |
| | | ✓ | 86.6 $\sigma$ 8.7 | 6.48 $\sigma$ 5.6 | 80.1 $\sigma$ 17 | 9.67 $\sigma$ 11 | 78.6 $\sigma$ 14 | 4.77 $\sigma$ 2.2 | 72.4 $\sigma$ 19 | 6.84 $\sigma$ 5.8 |
| | ✓ | ✓ | 87.6 $\sigma$ 7.1 | 6.35 $\sigma$ 5.2 | 80.4 $\sigma$ 17 | 10.6 $\sigma$ 12 | 78.0 $\sigma$ 15 | 5.15 $\sigma$ 2.4 | 71.1 $\sigma$ 20 | 7.85 $\sigma$ 8.0 |
| ✓ | | | **92.5** $\sigma$ 4.2 | **3.37** $\sigma$ 3.5 | 80.1 $\sigma$ 23 | 9.40 $\sigma$ 13 | 82.5 $\sigma$ 9.1 | 4.60 $\sigma$ 2.6 | 73.0 $\sigma$ 21 | 7.39 $\sigma$ 6.2 |
| ✓ | ✓ | | 92.4 $\sigma$ 4.0 | 3.49 $\sigma$ 3.7 | 84.2 $\sigma$ 17 | 7.90 $\sigma$ 11 | **83.2** $\sigma$ 8.6 | **4.35** $\sigma$ 2.0 | 75.8 $\sigma$ 18 | 6.78 $\sigma$ 6.5 |
| ✓ | | ✓ | 92.0 $\sigma$ 4.6 | 3.72 $\sigma$ 3.5 | 85.0 $\sigma$ 15 | 7.61 $\sigma$ 10 | 82.6 $\sigma$ 8.9 | 4.79 $\sigma$ 2.5 | 76.0 $\sigma$ 16 | 6.67 $\sigma$ 5.0 |
| ✓ | ✓ | ✓ | 92.1 $\sigma$ 4.4 | 3.60 $\sigma$ 3.8 | **86.2** $\sigma$ 13 | **7.02** $\sigma$ 9.5 | 82.9 $\sigma$ 8.9 | 4.29 $\sigma$ 2.2 | **77.0** $\sigma$ 16 | **6.33** $\sigma$ 5.8 |

dataset, we evaluated results on the original test set and the test set with the transformations described in Sec. 2.2. Note that we do not apply any of these corruptions to the training sets. Tab. 1 reports DSC and HD95 values for this experiment. Fig. 2 shows the relation between the severity of individual transformations and DSC values obtained for ten models.

**Cardiac cine MRI** For cardiac cine MRI, when not using any data augmentation, there is a large performance gap between the original ACDC test set and the transformed test set, i.e., DSC 0.891 vs. 0.755, indicating poor generalization to out-of-distribution data. We find that adding either MixUp or AFA to this model improves performance on the transformed test set, to DSC 0.760 and 0.801, respectively. Moreover, the combination of both augmentation strategies improves performance further to DSC 0.804. The gain on the transformed test set exceeds the performance drop on the original test set.

A similar pattern unfolds when we combine MixUp and AFA with base augmentations. Here, we see a performance increase in all settings for both the original and transformed data. Notably, base augmentations lead to a DSC of 0.801 on the transformed test set, compared to DSC 0.755 when no augmentation is used. This indicates that these augmentations are able to improve performance on some of the out-of-distribution samples. However, we find that MixUp and AFA can lead to significant ($p < 0.05$) performance gains on top of these augmentations, up to DSC 0.862 and HD95 7.02 when both are used. Moreover, when MixUp and AFA are used in combination with base augmentations, the performance drop on the original test set is smaller and not significant.

The results in Fig. 2 indicate that adding MixUp and AFA improves robustness to *all* imaging variations. This includes common MRI artifacts such as bias fields, which might be corrected using existing techniques, but also corruptions that can not easily be corrected with pre-processing techniques, such as k-space subsampling, ghosting, spike noise, and Rician noise. We also find that the performance increase is larger at higher severity levels again indicating the significant improvements to robustnes.

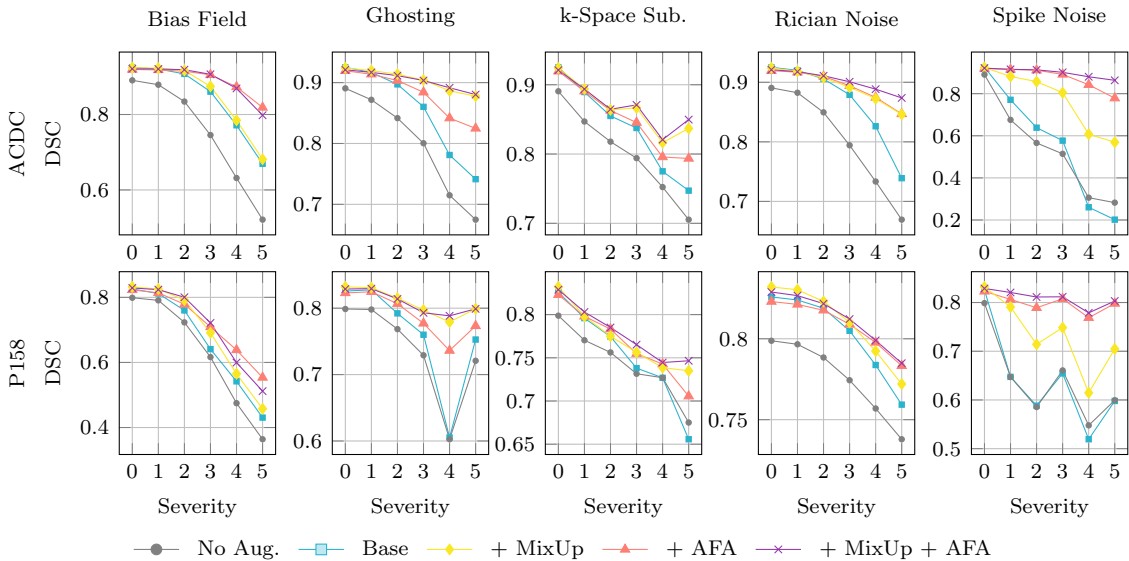

Figure 2: Trend of DSC per severity for test sets corrupted with bias field, ghosting, k-space subsampling, Rician noise and spike noise. We notice that MixUp and AFA are effective in mitigating the challenges posed by complex image corruptions.

**Prostate MRI**  Our findings in the prostate MRI data set match those found in the cardiac cine MRI data set to a large extent. We find that in the model without any augmentations, there is a performance gap between the original (DSC 0.789) and transformed (DSC 0.705) data sets. Using AFA, either stand-alone or in combination with MixUp, narrows this gap. However, we also find that using *only* MixUp has a detrimental effect on model performance. Similar to the cardiac cine MRI set, we find that MixUp and AFA reduce performance on the original data set when used without base augmentations; however, they only significantly reduce when using both MixUp and AFA. In contrast to cardiac cine MRI, adding MixUp and AFA to base augmentations not only leads to a significant performance increase on the transformed data set but also on the original dataset. Results for prostate MRI in Fig. 2 show similar trends as for cardiac cine MRI, confirming the general nature of our findings and added value of MixUp and AFA over base augmentations.

### 3.2. Real-World Distribution Shifts

In our previous experiments, we analysed the robustness of models trained with various augmentation s on out-of-distribution test sets where the distribution shift was controlled. We now consider generalization between datasets with real-world distribution shifts. These may be a result of different demographics, protocols, acquisition parameters like resolution, b-values in prostate bi-parametric MR, scanner vendors, etc. As in our previous experiment, we train with combinations of base, MixUp and AFA augmentation, omitting results trained without base augmentations. Tab. 2 lists results for cardiac cine MRI and prostate MRI.

Table 2: DSC and HD95 performance under distribution shift for Cardiac Cine MR, testing on M&Ms, and Prostate bpMRI, testing on PX, with various data augmentation strategies. Blue numbers denote significantly improved metrics when models using MixUp or AFA compared to model only using base augmentation ($p < 0.05$).

| Augmentation | | | Cardiac Cine MR | | | Prostate bpMRI | | |
|---|---|---|---|---|---|---|---|---|
| Base | MixUp | AFA | Trained On | DSC | HD95 (mm) | Trained On | DSC | HD95 (mm) |
| ✓ | | | ACDC | $87.0 \ _{\sigma\,8.3}$ | $7.97 \ _{\sigma\,9.3}$ | P158 | $70.5 \ _{\sigma\,21}$ | $7.87 \ _{\sigma\,7.4}$ |
| ✓ | ✓ | | | $88.0 \ _{\sigma\,7.5}$ | $5.74 \ _{\sigma\,5.6}$ | | $73.7 \ _{\sigma\,17}$ | $6.60 \ _{\sigma\,4.4}$ |
| ✓ | | ✓ | | $87.4 \ _{\sigma\,7.6}$ | $6.92 \ _{\sigma\,7.2}$ | | $71.8 \ _{\sigma\,18}$ | $7.67 \ _{\sigma\,9.5}$ |
| ✓ | ✓ | ✓ | | $88.0 \ _{\sigma\,7.2}$ | $5.70 \ _{\sigma\,5.3}$ | | $73.2 \ _{\sigma\,17}$ | $7.52 \ _{\sigma\,6.7}$ |
| | | | M&Ms[†] | $88.2 \ _{\sigma\,6.7}$ | $5.02 \ _{\sigma\,4.6}$ | PX[+] | $82.6 \ _{\sigma\,9.0}$ | $4.77 \ _{\sigma\,3.5}$ |

Best model trained on the test dataset from † (Campello et al., 2021) and + (Xu et al., 2023).

**Cardiac cine MR**  We train segmentation models on the ACDC data set and use M&Ms as a test set. We find that a model trained on ACDC with only base augmentations experiences a performance drop compared to a model trained on M&Ms with only base augmentations, with Dice coefficients of 0.870 and 0.882. However, when combinations of MixUp with AFA are added, we find consistent performance improvements across all augmentation combinations. Furthermore, we find that for models that use a combination of base augmentations, MixUp, or both MixUp and AFA, we approach the DSC and HD95 score of the model that was trained on M&Ms itself, bridging the generalization gap.

**Prostate MRI**  The segmentation of the prostate bpMRI presented a more significant domain shift challenge. We train the segmentation model on P158 and use the PX dataset as the test set. As noted earlier, the large variability in prostate glands poses a difficult challenge and substantially impacts model performance. Despite this challenge, a model trained with base augmentations, along with MixUp (DSC 0.737, HD95 6.60), is a significant improvement over using only the base augmentations (DSC 0.705, HD95 7.87), indicating improved generalization capabilities in this setting where there is large variance in anatomy. Models in combination with AFA also showcase significantly improved performance.

### 3.3. Model Interpretation

To further analyse why the proposed augmentations outperform the base augmentation in MRI, we quantify the separability and compactness of their learned features using k-variance gradient-normalized margins (kVGM) (Chuang et al., 2021). A higher value for this metric indicates that the model has learned more separable and compact clusters of representations which in turn is linked to better generalisabilty. Fig. 3 visualizes the position of voxels from the transformed test set in the feature space of ten of our trained models (dimensionality reduced via PCA), along with the kVGM of each model. These plots show that the absence of augmentation leads to poor feature separation, while using only base augmentations leads to better clustering of features that are not easily separable. Adding AFA alone improves separability, and MixUp alone enhances compactness, and when combined, they appear to promote both compactness and separability. In Appendix E we show that this behaviour

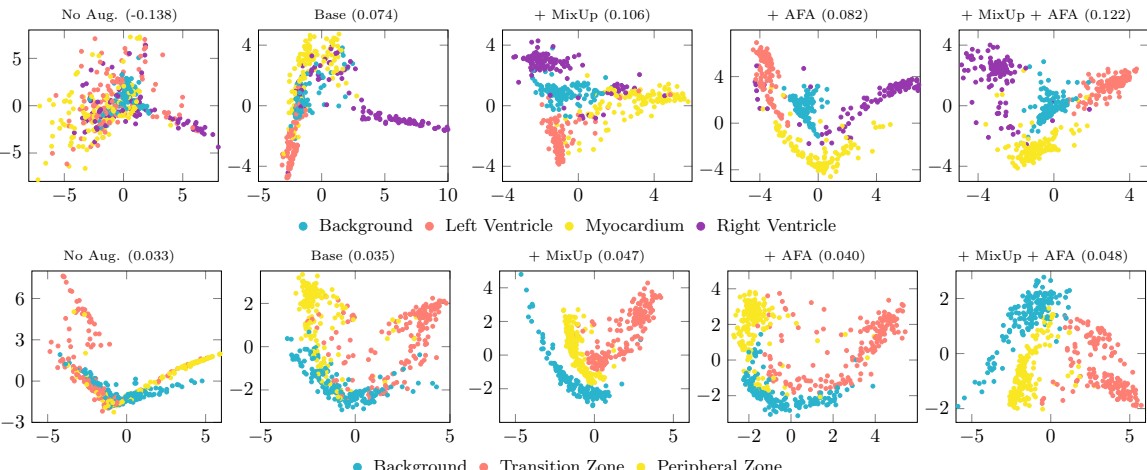

Figure 3: PCA projection of learned features, for final features from nnU-Net trained with different augmentation techniques for samples from the transformed test sets (top: ACDC, bottom: P158) with the corresponding kVGM metric.

is consistent across many runs. The kVGM metric, in increasing order for generalisability, ranks the models starting from no augmentations, followed by base augmentations, base with AFA, base with MixUp, and finally base with both AFA and MixUp. This supports our finding that these augmentations enrich feature representations, leading to enhanced out-of-distribution generalization.

## 4. Discussion and Conclusion

In this study, we have demonstrated how non-standard augmentation techniques that do not target specific variations, specifically MixUp and Auxiliary Fourier Augmentation (AFA), can enhance the robustness of state-of-the-art segmentation frameworks like nnU-Net against many variations in MRI.

While MixUp has been known to be an effective augmentation for various tasks (Eaton-Rosen et al., 2018; Thulasidasan et al., 2019; Gazda et al., 2022), we find it is very well suited for overcoming challenging medical image conditions as well. However, we also observe that without base augmentations on P158, MixUp alone leads to a (non-significant) decline in performance. Our results highlight the advantage of combining augmentation strategies that intrinsically exploit different mechanisms. For example, AFA follows a fundamentally different strategy than MixUp by directly perturbing $k$-space data. The effect of this is shown in our results, in which the combination of both always improves over their individual use. This is corroborated by an evaluation of the feature space using $k$-variance gradient-normalized margins. We consider this metric a promising tool for studying model generalizability.

Our results demonstrate that MixUp and AFA not only improve robustness to distribution shifts but also maintain comparable performance on the original, non-transformed dataset. This is a significant advantage, as many robustness techniques, such as adversar-

ial training or aggressive noise injection, often introduce biases that degrade performance on standard tasks (Tsipras et al., 2019; Hendrycks et al., 2019; Zhang et al., 2019; Geirhos et al., 2020). These biases arise because such methods overfit to the augmented or corrupted data. In contrast, our considered augmentations promote feature compactness and separability without disrupting the underlying data distribution, ensuring that performance on the original dataset remains consistent and comparable. This balance between robustness and accuracy is critical for clinical applications, where models must perform reliably across both clean and challenging data.

Our results align with studies incorporating general augmentation strategies into nnU-Net (Atya et al., 2021), and MixUp and AFA are straightforward additions with a lot of benefits. However, augmentations cannot address all generalization gaps. We find that prostate zonal segmentation remains challenging due to significant inter-subject variability. One example is the effect of age, where younger cohorts have sharper tissue boundaries (Allen et al., 1989; Situmorang et al., 2012) and such differences are difficult to address without prior knowledge, regardless of augmentations.

While the augmentations we treat in this work are valuable tools for improving robustness, there remains substantial potential for further advancements in this domain. For example, we have here used a base version of MixUp which has been previously shown to be sufficient (Atya et al., 2021), but there are many variants (Cao et al., 2024). We further include discussion on CutMix in Appendix G and how it pairs with other augmentations. However, most other variants would involve too many changes to the nnU-Net framework for limited benefits (Liu et al., 2024) and some are superfluous to analyse as, for instance, nnU-Net uses deep supervision (Shen et al., 2020) and therefore the use of base MixUp is similar to Manifold-MixUp (Verma et al., 2019). There are other data-agnostic augmentations as well which are out of scope of discussion, and would need to be adapted for medical domain, like PRIME (Modas et al., 2022), which considers only RGB color space which is characteristically different from multi-parametric MRI scans and is not implemented for 3D volumes. Therefore, our work can be viewed as a stepping stone toward broader research on using simple general augmentations for out-of-distribution generalization in medical imaging, as opposed to more complicated methods like model-based methods like GANs and diffusion models (Garcea et al., 2023).

In conclusion, we find that adding non-standard data-agnostic augmentation to a state-of-the-art nnU-Net model can consistently and significantly increase segmentation performance under various generalisation challenges, for cardiac cine MRI and prostate MRI. This could enhance the reliability of segmentation models under diverse and challenging conditions in clinical practice.

## Acknowledgments

This publication is part of the project ROBUST: Trustworthy AI-based Systems for Sustainable Growth with project number KICH3.LTP.20.006, which is (partly) financed by the Dutch Research Council (NWO), Siemens Healthineers, and the Dutch Ministry of Economic Affairs and Climate Policy (EZK) under the program LTP KIC 2020-2023.

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

## Appendix A. More Examples of Variations

We show an example of all the image variations that we make part of the transformations. In Fig. 4 we show an example image from each transformed test set for ACDC and in Fig. 5 we show an example image from each transformed test set of P158.

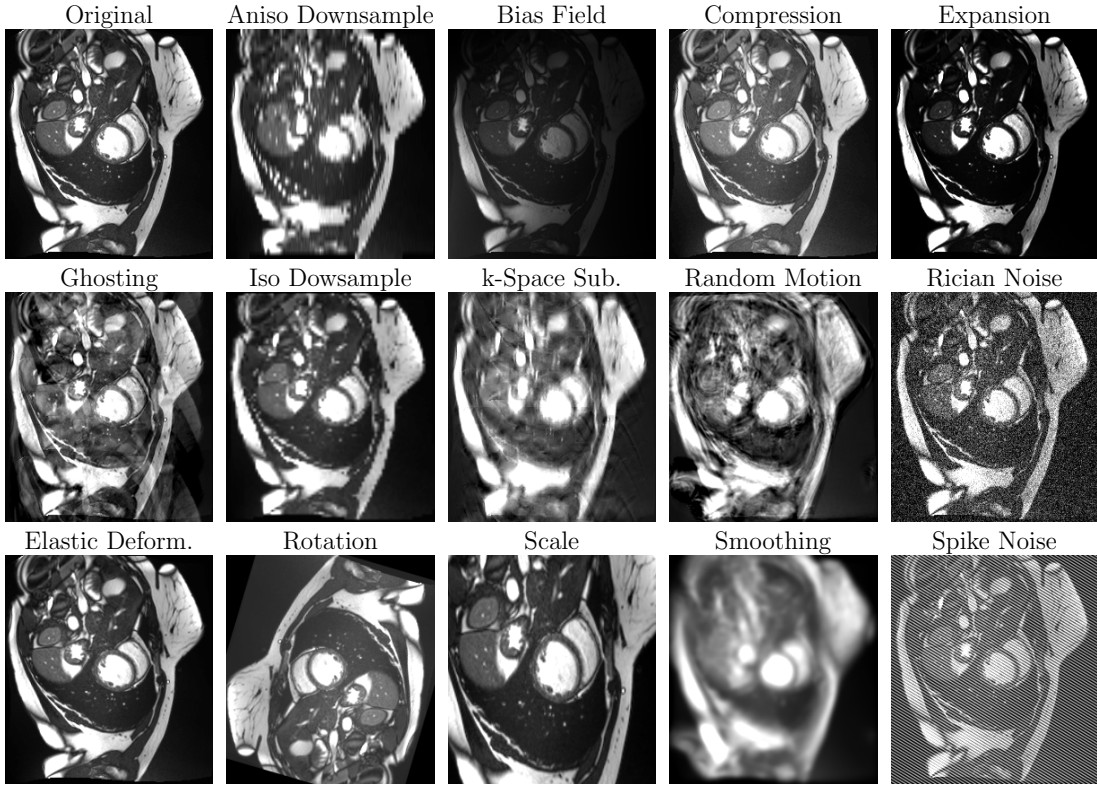

Figure 4: Visualization of the 14 data variations, alongside the original image (top-left) for a test sample in the ACDC dataset. All transforms visualised at severity 3.

The images show while the object of interest remains discernable to the human eye, the difference to the original sample is large. Some variations are not diagnostically relevant, like smoothing, severe random motion, but they serve as interesting examples of where human expertise might still outperform sophisticated deep learning methods, and through this study we explore how to bridge this gap to unknown variations without explicitly using them as augmentations.

## Appendix B. Example Images of Augmentation

In this section we show some examples of images as a result of applying the augmentation strategies. In Fig. 6 we show an example of AFA augmentation and in Fig. 7 we show the result of a mixup augmentation on both the image and the segmentation masks.

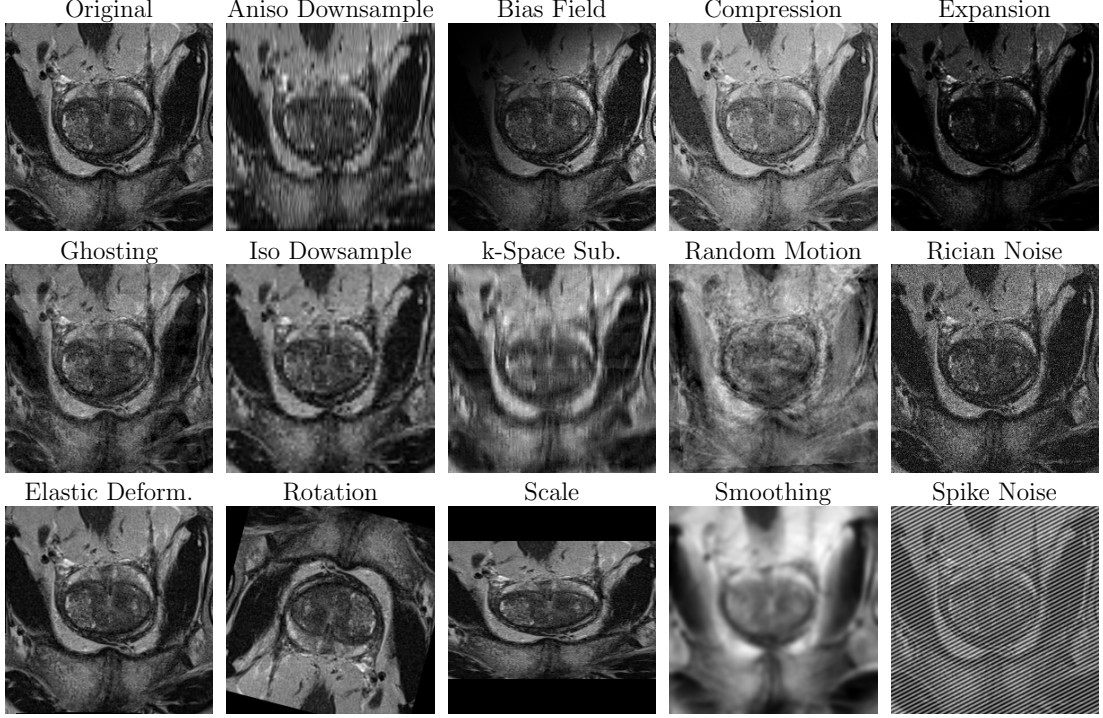

Figure 5: Visualization of the 14 data variations, alongside the original image (top-left) for a test sample in the P158 dataset. All transforms visualised at severity 3.

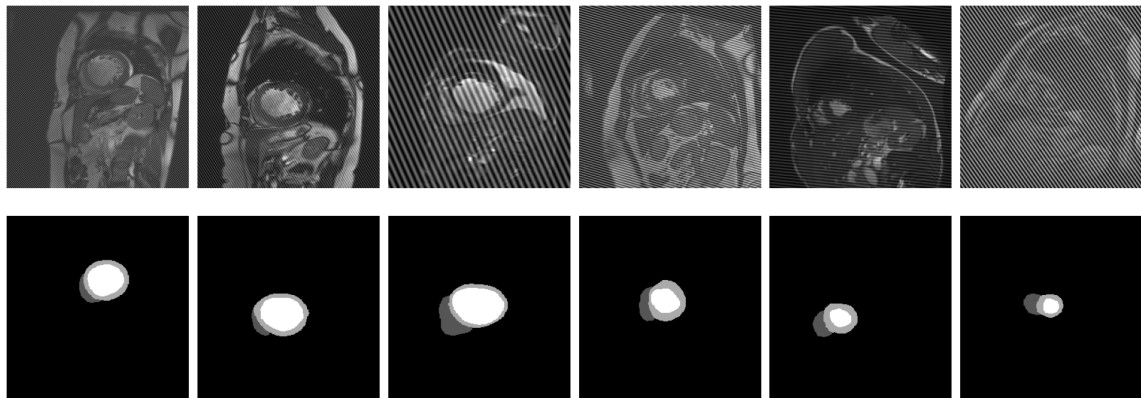

Figure 6: These images have been augmented using AFA. The resultant image has varying degree and ampltide of planar waves if done on a 2D slice. The labels are left unaffeted.

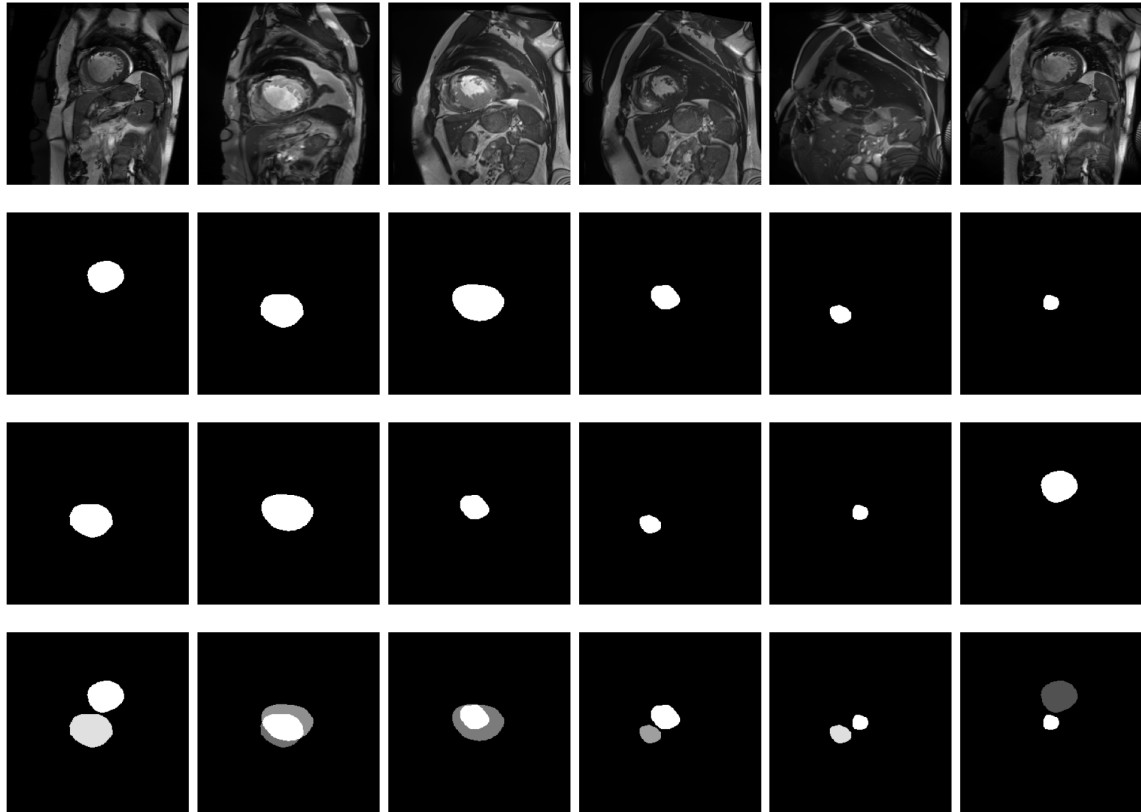

Figure 7: This image shows the effects of a mixup augmentation on both the images and the labels. The first row shows examples of samples that have mixup applied to them. Row 2 and Row 3 are the ground truth labels for myocardium of the samples before being mixed up. We omit showing the other classes for visualisation. In the last row we show how the effect on the class and how the segmentation masks are combined to produce a probaility mask which is then used as a ground truth during loss calculation.

## Appendix C. Hyperparameters and Reproducibility

In this section, we give an exhaustive overview of the hyperparameters used for training of the several nnU-Nets for the various datasets. We make our code available at https://github.com/MIAGroupUT/augmentations-for-the-unknown.

**nnU-Net Hyperparameters** We summarise our used hyperparameters for training all the nnU-Net architectures unless specified otherwise in Tab. 3.

**Computational Cost and Convergence** MixUp incurs no additional computational overhead, requiring $1\times$ FLOPs and memory compared to the baseline. In contrast, AFA is slightly more expensive, with $2\times$ FLOPs and $1.62\times$ memory usage. This increase is due

Table 3: Default nnU-Net hyperparameters and modifications used in our experiments.

| Category | Details |
|---|---|
| **Architecture** | |
| - Model Type | 2D U-Net (default); full-resolution 3D U-Net for brain MRI. |
| - Ensembles | Predictions ensembled across 5 models (five-fold cross-validation). |
| **Training** | |
| - Optimizer | SGD with Nesterov momentum (momentum = 0.99). |
| - Learning Rate | Initial rate = 0.01, polynomial decay (power = 0.9). |
| - Regularization | Weight decay = 3e-5. |
| - Epochs | Maximum of 200 (modified). |
| - Batch Size | Adaptive to GPU memory (2-5 for 3D, larger for 2D). |
| - Loss Function | Combined Dice Loss and Cross-Entropy Loss. |
| **Data Preprocessing** | |
| - Intensity Normalization | Z-score normalization (per channel). |
| - Resampling | Voxel spacing resampled to median value. |
| - Padding | Mirror padding applied. |
| **Inference** | |
| - Test-Time Augmentation (TTA) | Mirroring along all axes. |
| **Post-Processing** | |
| - Connected Component Analysis | Applied for class-specific refinement. |

to the additional Fourier-based transformations and auxiliary losses, which are designed to enhance robustness without significantly impacting efficiency. Despite the higher computational cost of AFA, the convergence speed remains unaffected. All models, including AFA and MixUp, fully converge by 200 epochs. This demonstrates that the added complexity does not delay training, ensuring that the benefits of improved robustness and generalization are achieved without sacrificing training efficiency.

## Appendix D. Performance per Severity for All Corruptions

We plot the trend of DSC metric per severity for each transformation considered under image variations. Trends for DSC for ACDC are shown in Fig. 8 and HD95 in Fig. 9. Trends for DSC for P158 are shown in Fig. 10 and HD95 in Fig. 11.

We see that while base augmentations are really effective in improving generalisation to some transformations, rotation, scale contrast compression and expansion, and iso-downsample. These are transformations that base augmentations also overlap with. However, more complicated transformations are not completely overcome, for instance, bias field, ghosting, rician noise, k-space subsampling, spike noise, smoothing, aniso-downsampling and random motion.

The augmentations MixUp and AFA also seem to complement each other. Using AFA without MixUp can reduce performance on HD95 metric on some transformations, which can be understood by the fact that regularising frequency components leads to difficulty

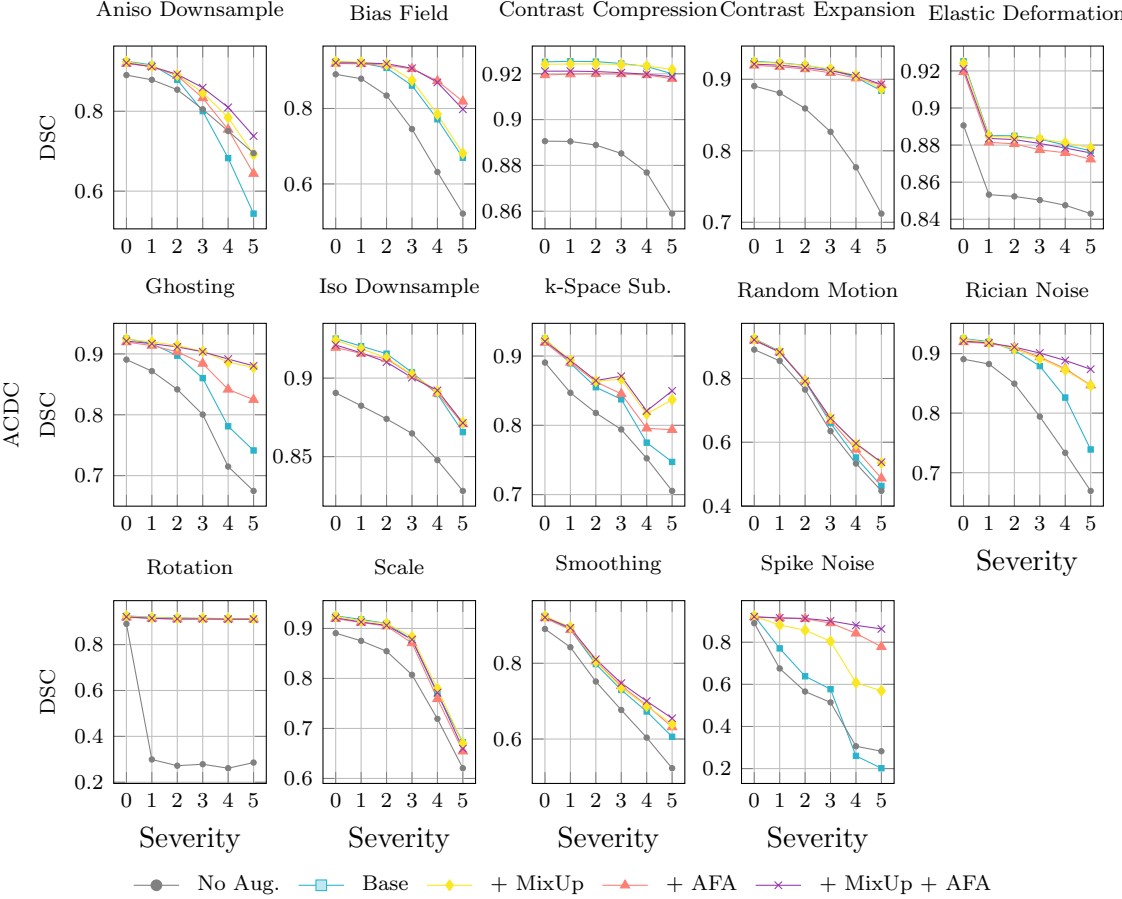

Figure 8: Trend of DSC per severity for test sets transformed with 14 different transformations for the ACDC dataset, including from the 5 repeated from the main paper.

delineating boundaries (typical high frequency changes) while MixUp does not deteriorate boundary delineation, but it is unable to regularise frequency components leading to performance decline on various variations. However, using both MixUp and AFA in general leads to the best performance for each metric, and this pattern is consistent between both test datasets.

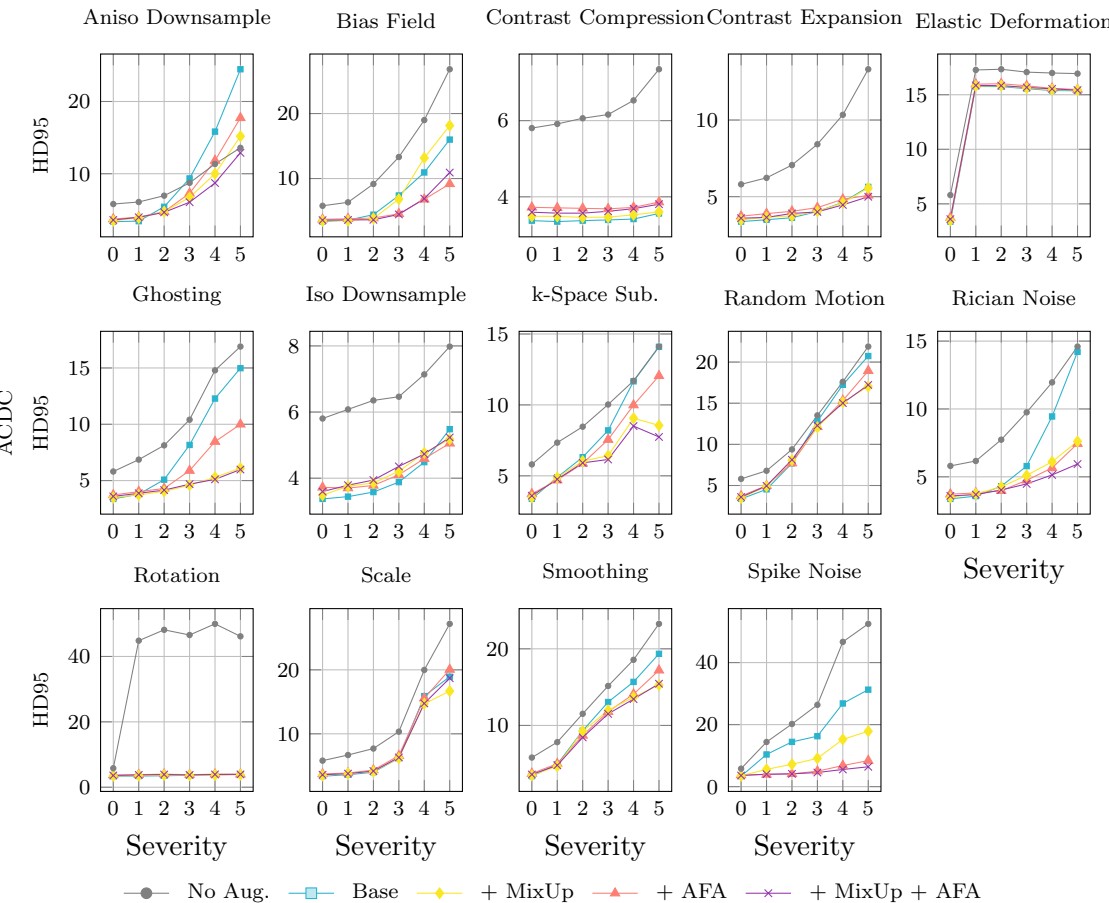

Figure 9: Trend of HD95 per severity for test sets transformed with 14 different transformations for the ACDC dataset.

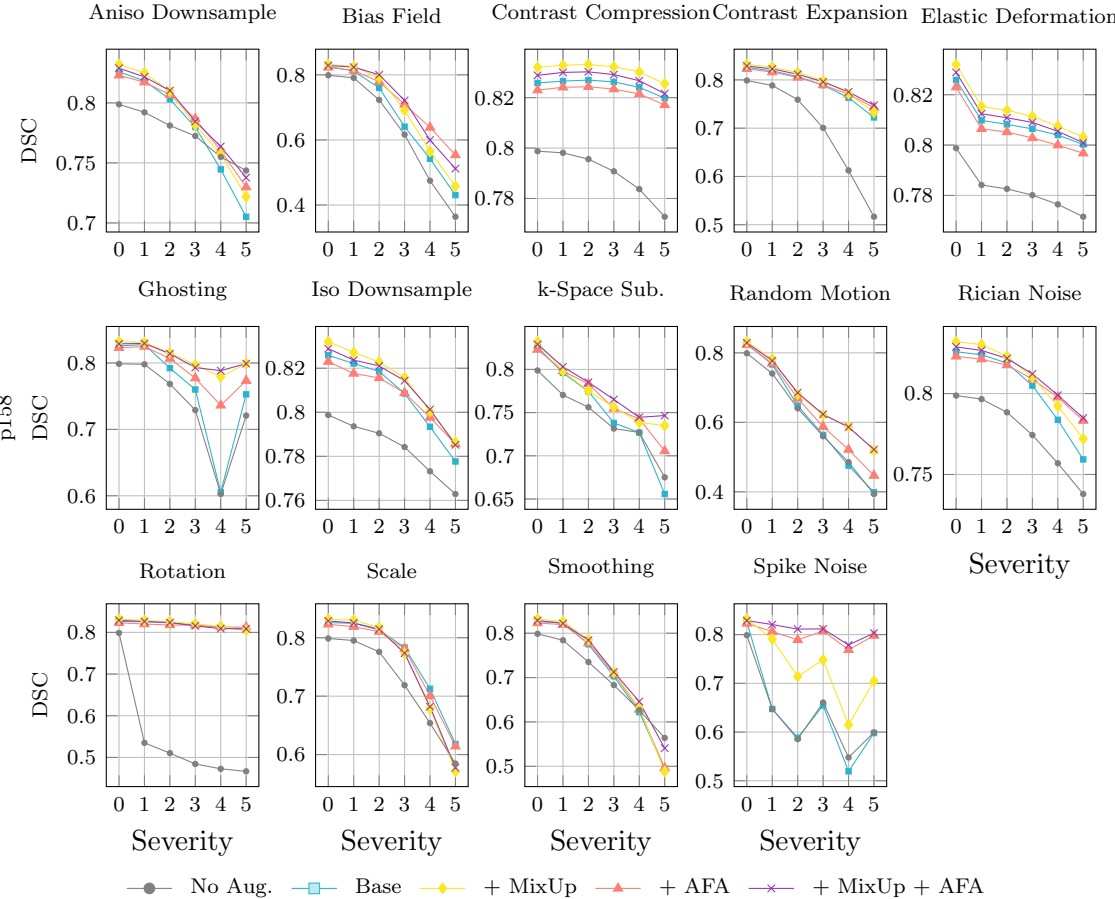

Figure 10: Trend of DSC per severity for test sets transformed with 14 different transformations for the P158 dataset, including the 5 presented from the main paper.

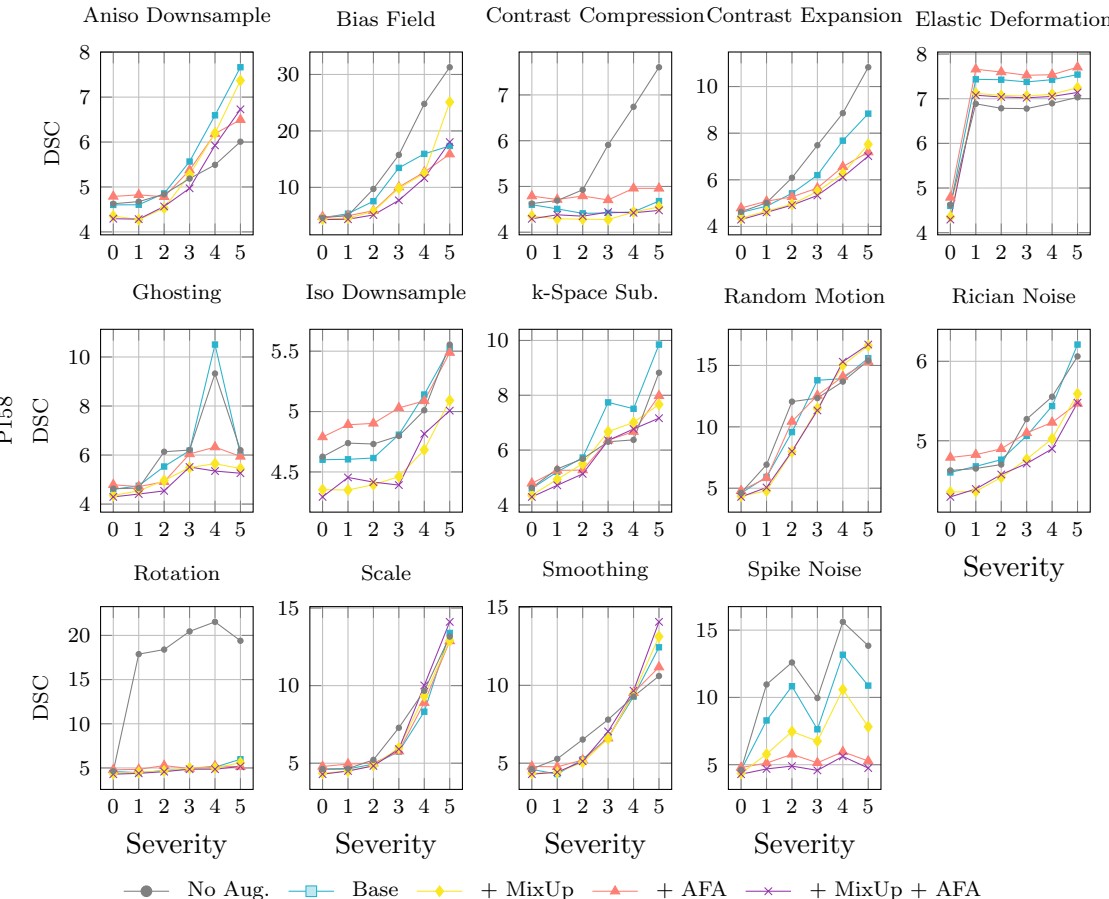

Figure 11: Trend of DSC per severity for test sets transformed with 14 different transformations for the P158 dataset.

## Appendix E. Latent Space Representations across Initialisations

Here we repeat the PCA projection of the learned features for the final features from nnU-Net trained with different augmentation techniques across different runs. We take the model trained in each fold of our 5-cross validation training process for this purpose. The latent space representation plots for ACDC are shown in Fig. 12 and for P158 in Fig. 13. Both from qualitative and quantitative perspective these repetitions show consistently similar learnt feature representation under images with transformations.

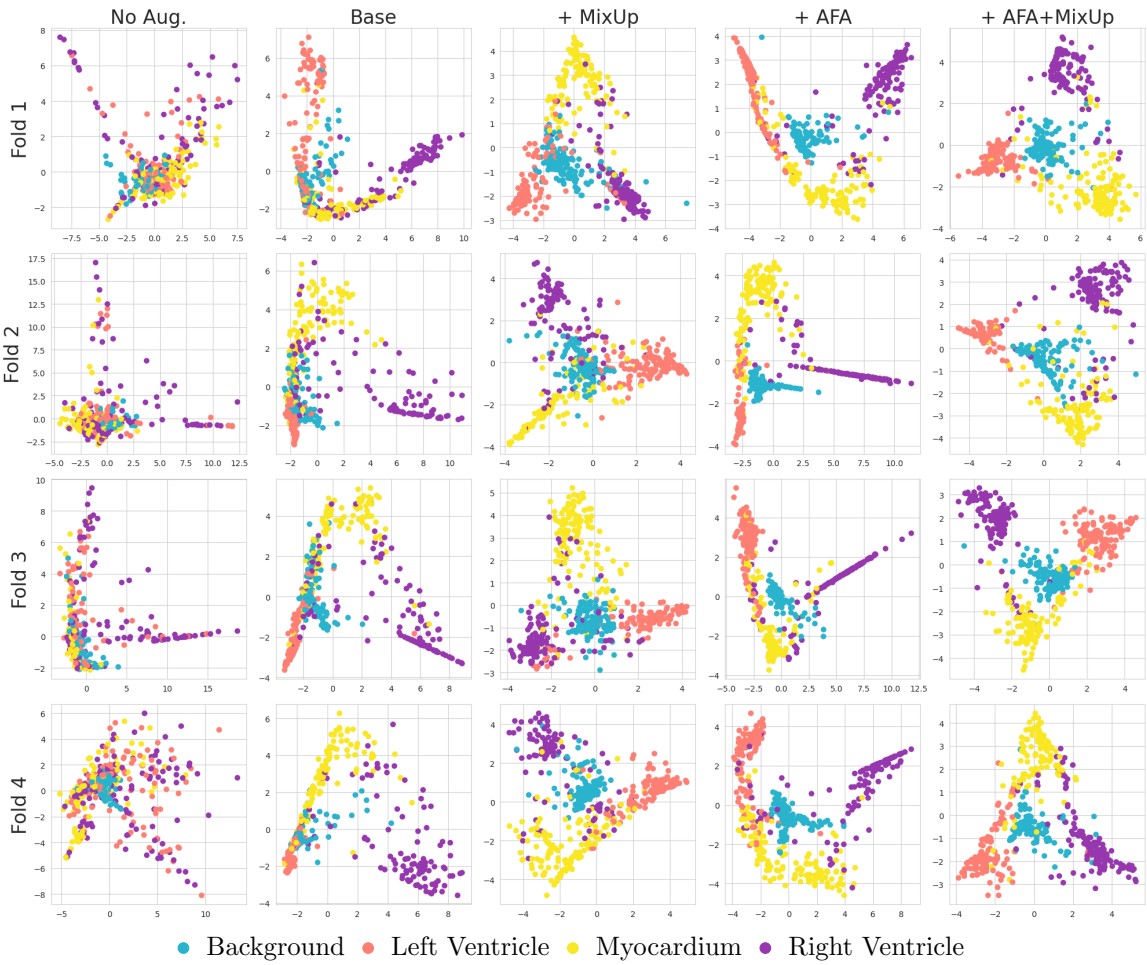

Figure 12: We perform more iterations of our analysis of the learnt feature representation for ACDC dataset over different folds of our five-fold cross validation training. The one wirtten in the paper is from fold 0, and so here we show the rest 4 fold of the models trained with image augmentations: none, only base augmentation and in combination with MixUp or AFA or both.

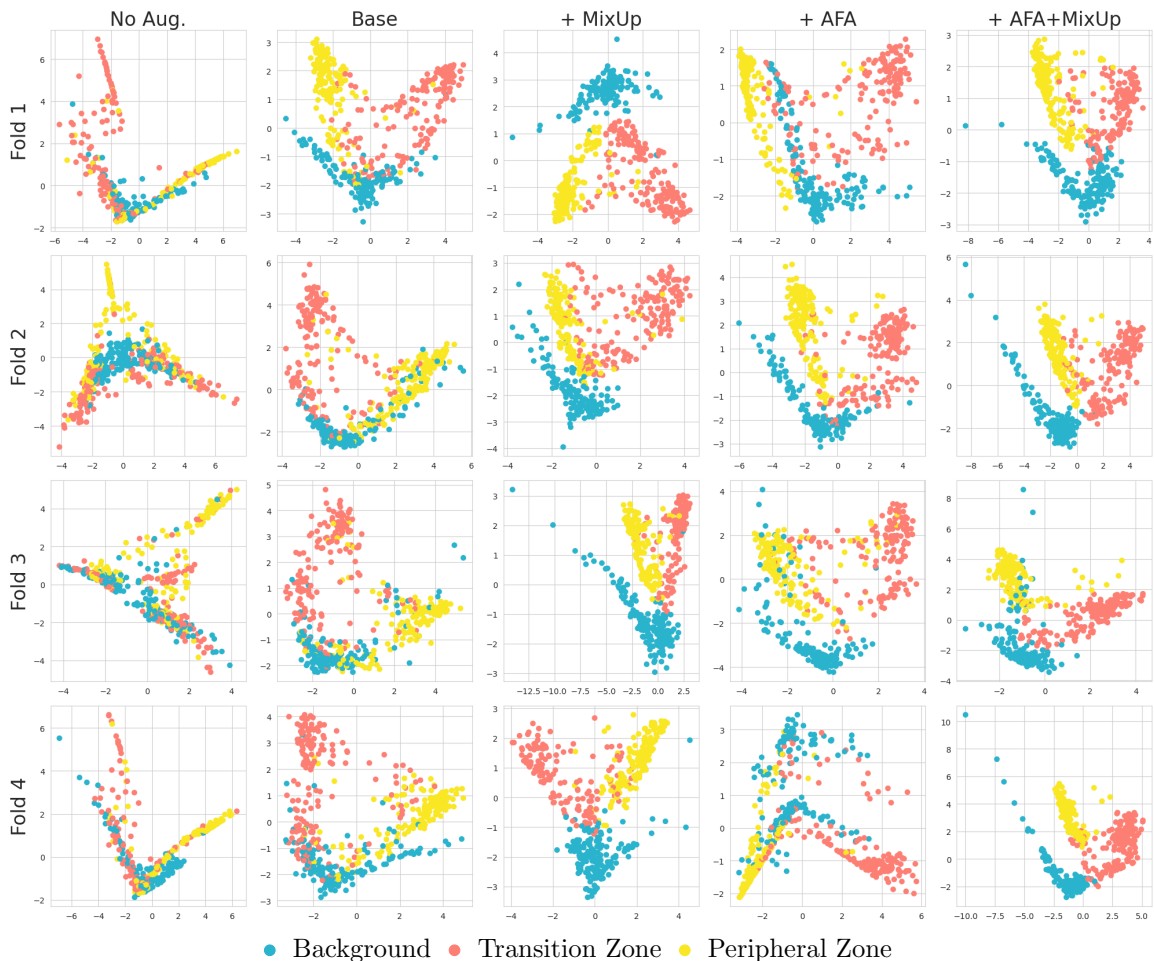

Figure 13: We perform more iterations of our analysis of the learnt feature representation for P158 dataset over different folds of our five-fold cross validation training. The one wirtten in the paper is from fold 0, and so here we show the rest 4 fold of the models trained with image augmentations: none, only base augmentation and in combination with MixUp or AFA or both.

## Appendix F. Evidence of Regularisation

L2 regularisation is often used in training deep neural networks to reduce overfitting to noise by promoting learnt weights to have a lower l2-norm. We see that while we do not explicitly regularise the models using l2-norm, base augmentations and MixUp lead to substantially lower norms of the learnt convolutional kernels on models trained for both cardiac cine MR (Fig. 14) and prostate bpMRI (Fig. 15).

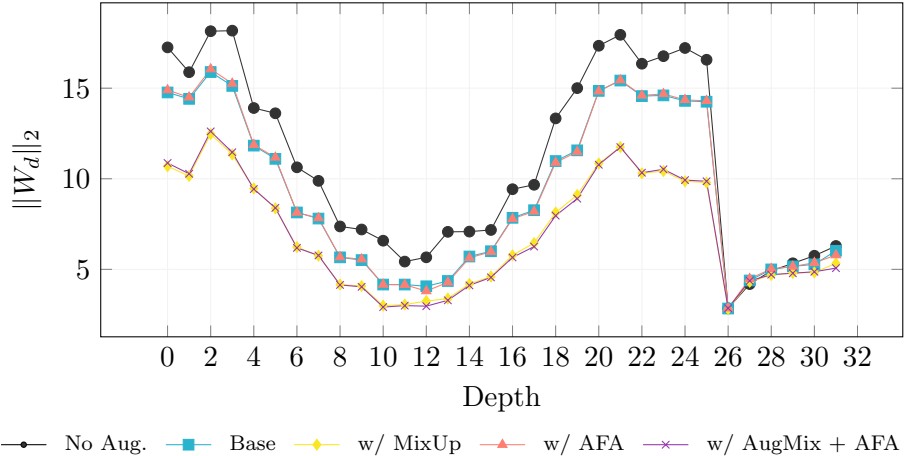

Figure 14: The norm of the weights of convolutional kernels $W$ at different depths, $d$, for different nnU-Nets trained on ACDC with different augmentation techniques. The plot highlights the regularisation effect the methods have on the model weights.

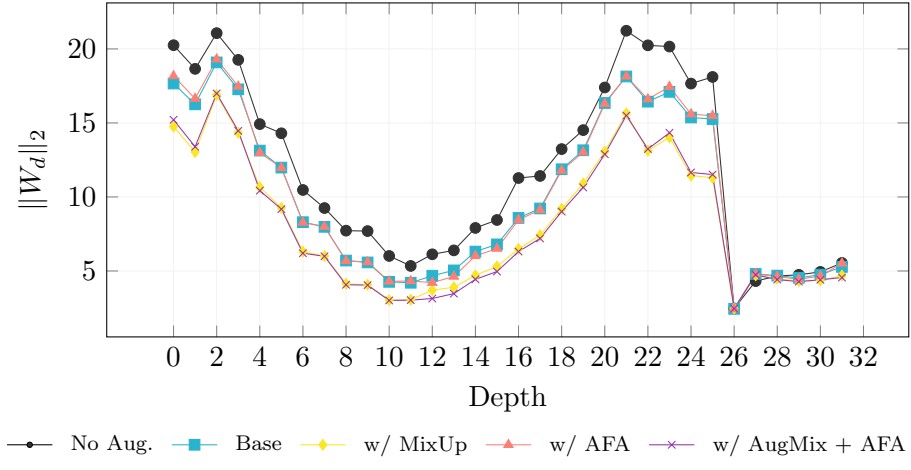

Figure 15: The norm of the weights of convolutional kernels $W$ at different depths, $d$, for different nnU-Nets trained on P158 with different augmentation techniques. The plot highlights the regularisation effect the methods have on the model weights.

## Appendix G. Additional Experiments

We have conducted additional experiments with CutMix (Yun et al., 2019) and one additional dataset for Brain MRI using the Alzheimer's Disease Neuroimaging Initiative Hippocampus Segmentation (ADNI) dataset (Frisoni et al., 2015) and an additional OOD distribution dataset Hippocampus Segmentation (HFH) (Jafari-Khouzani et al., 2011) to

evaluate the robustness of our augmentations to image transformations. We present these preliminary results for robustness to image variation in Tab. 4 and results for real world out-of-distribution in Tab. 5 and include the results from the manuscript again for completeness.

Table 4: DSC and HD95 on the original and transformed test set of ACDC, P158 and ADNI using either using no augmentations or a combination of base, MixUp, CutMix, and AFA augmentations.

| Augmentation | | | | ACDC | | | | P158 | | | | ADNI | | | |
| | | | | Original | | Transformed | | Original | | Transformed | | Original | | Transformed | |
| Base | MixUp | CutMix | AFA | DSC | HD95 (mm) | DSC | HD95 (mm) | DSC | HD95 (mm) | DSC | HD95 (mm) | DSC | HD95 (mm) | DSC | HD95 (mm) |
|---|---|---|---|---|---|---|---|---|---|---|---|---|---|---|---|
| ✓ | | | | 92.5 | 3.37 | 80.1 | 9.40 | 82.5 | 4.60 | 73.0 | 7.39 | 90.1 | 1.07 | 77.9 | 3.45 |
| ✓ | ✓ | | | 92.4 | 3.49 | 84.2 | 7.90 | 83.2 | 4.35 | 75.8 | 6.78 | 90.2 | 1.08 | 82.4 | 2.30 |
| ✓ | | ✓ | | 93.0 | 3.35 | 79.9 | 9.69 | 84.0 | 4.64 | 71.8 | 8.14 | 90.1 | 1.08 | 78.0 | 3.45 |
| ✓ | | | ✓ | 92.0 | 3.72 | 85.0 | 7.61 | 82.6 | 4.79 | 76.0 | 6.67 | 90.2 | 1.08 | 80.3 | 2.43 |
| ✓ | ✓ | | ✓ | 92.1 | 3.60 | 86.2 | 7.02 | 82.9 | 4.29 | 77.0 | 6.33 | 90.1 | 1.07 | 83.0 | 1.94 |
| ✓ | ✓ | ✓ | ✓ | 92.6 | 3.47 | 86.2 | 7.04 | 83.9 | 4.54 | 77.5 | 6.28 | 90.0 | 1.07 | 81.7 | 2.60 |

Table 5: DSC and HD95 performance under distribution shift for Cardiac Cine MR, testing on M&Ms, Prostate bpMRI, testing on PX, and Brain MR, testing on HFH, with various data augmentation strategies. HFH does not have a public leaderboard for best model performance.

| Augmentation | | | | Cardiac Cine MR | | | Prostate bpMRI | | | Brain MR | | |
| Base | MixUp | CutMix | AFA | Trained On | DSC | HD95 (mm) | Trained On | DSC | HD95 (mm) | Trained On | DSC | HD95 (mm) |
|---|---|---|---|---|---|---|---|---|---|---|---|---|
| ✓ | | | | | 87.0 | 7.97 | | 70.5 | 7.87 | | 78.0 | 4.93 |
| ✓ | ✓ | | | | 88.0 | 5.74 | | 73.7 | 6.60 | | 79.2 | 2.39 |
| ✓ | | ✓ | | ACDC | 88.3 | 5.64 | P158 | 77.2 | 5.64 | ADNI | 79.4 | 2.37 |
| ✓ | | | ✓ | | 87.4 | 6.92 | | 71.8 | 7.67 | | 79.0 | 2.42 |
| ✓ | ✓ | | ✓ | | 88.0 | 5.70 | | 73.2 | 7.52 | | 79.0 | 2.44 |
| ✓ | ✓ | ✓ | ✓ | | 88.0 | 6.91 | | 77.1 | 6.23 | | 79.2 | 2.39 |
| | | | | M&Ms | 88.2 | 5.02 | PX | 82.6 | 4.77 | HFH | - | - |

Best model trained on the test dataset from † (Campello et al., 2021) and + (Xu et al., 2023).

We found that CutMix provides better generalisation performance both on the original and the real world OOD test set. However, we found that the performance of CutMix is significantly worse on the test set with various image transformations compared to MixUp and Auxiliary Fourier Augmentation. This is expected as CutMix, by design, learns better local and global feature representation by replacing patches and volumes within an MRI scan. However, it fails to produce diverse samples like MixUp and Auxiliary Fourier Augmentation, and therefore does not regularise the model for image transformations. However, the combination of MixUp, CutMix and AFA provides the best of both worlds and provides the out-of-distribution generalisation to image variations and real world datasets.

Most notably, the large real world OOD generalisation gap Prostate bpMRI is significantly reduced (from DSC of 0.737 to 0.772). This is now only 5.5% lower than training on the ProstateX dataset directly as opposed to the 8.9% lower when using MixUp with base augmentations only.

## Appendix H. Structure Wise Results and Standard Deviation of Metrics

In this section we report the structure wise means and the standard deviations of the metrics across the test set for the experiments reported in Tab. 1 and Tab. 2. In Tab. 6 we report the structure wise mean results of the DSC and HD95 metrics for experiments where we train the model on ACDC and test on the transformed test set. In Tab. 7 reports the standard deviations of the same metrics. We average over the phase for these table due to compounding size of the tables.

Table 6: DSC and HD95 on the original and transformed test set of ACDC and P158 using either using no augmentations or a combination of base, MixUp, CutMix, and AFA augmentations. We do not stratify over the phase (and is averaged over) here due to the compounding size of the table.

| Augmentation | | | | Original | | | | | | Transformed | | | | | |
| | | | | LV | | MYO | | RV | | LV | | MYO | | RV | |
| Base | MixUp | CutMix | AFA | DSC | HD95 | DSC | HD95 | DSC | HD95 | DSC | HD95 | DSC | HD95 | DSC | HD95 |
|---|---|---|---|---|---|---|---|---|---|---|---|---|---|---|---|
| | | | | 88.0 | 7.06 | 87.7 | 4.72 | 91.5 | 5.63 | 66.6 | 20.61 | 70.0 | 11.74 | 80.3 | 11.84 |
| | ✓ | | | 88.0 | 6.98 | 87.3 | 5.37 | 91.4 | 5.89 | 67.8 | 17.98 | 69.1 | 13.80 | 80.6 | 11.77 |
| | | | ✓ | 85.8 | 7.94 | 84.3 | 5.59 | 89.9 | 5.93 | 72.0 | 15.76 | 71.8 | 11.32 | 80.9 | 10.36 |
| | ✓ | | ✓ | 87.0 | 7.19 | 85.6 | 5.74 | 90.3 | 6.11 | 72.3 | 16.46 | 72.1 | 11.83 | 82.9 | 10.65 |
| ✓ | | | | 92.1 | 4.30 | 90.9 | 2.85 | 94.6 | 2.97 | 78.5 | 12.84 | 78.4 | 7.21 | 86.6 | 6.88 |
| ✓ | ✓ | | | 92.1 | 4.26 | 91.0 | 2.95 | 94.2 | 3.26 | 83.3 | 10.96 | 81.9 | 5.99 | 88.9 | 6.04 |
| ✓ | | ✓ | | 92.8 | 4.14 | 91.6 | 2.69 | 94.5 | 3.23 | 79.6 | 12.45 | 78.6 | 7.50 | 85.4 | 7.37 |
| ✓ | | | ✓ | 91.6 | 4.32 | 90.4 | 3.10 | 93.9 | 3.75 | 84.0 | 10.72 | 82.0 | 5.82 | 89.5 | 5.93 |
| ✓ | ✓ | | ✓ | 91.7 | 4.46 | 90.6 | 3.04 | 94.0 | 3.29 | 85.5 | 9.94 | 83.4 | 5.30 | 90.0 | 5.61 |
| ✓ | ✓ | ✓ | ✓ | 92.0 | 5.98 | 91.3 | 2.97 | 94.4 | 3.43 | 85.8 | 11.60 | 83.8 | 5.46 | 90.4 | 5.65 |

Table 7: The standard deviation of DSC and HD95 on the original and transformed test set of ACDC and P158 using either using no augmentations or a combination of base, MixUp, CutMix, and AFA augmentations. We do not stratify over the phase (and is averaged over) here due to the compounding size of the table.

| Augmentation | | | | Original | | | | | | Transformed | | | | | |
| | | | | LV | | MYO | | RV | | LV | | MYO | | RV | |
| Base | MixUp | CutMix | AFA | $\sigma_{\text{DSC}}$ | $\sigma_{\text{HD95}}$ | $\sigma_{\text{DSC}}$ | $\sigma_{\text{HD95}}$ | $\sigma_{\text{DSC}}$ | $\sigma_{\text{HD95}}$ | $\sigma_{\text{DSC}}$ | $\sigma_{\text{HD95}}$ | $\sigma_{\text{DSC}}$ | $\sigma_{\text{HD95}}$ | $\sigma_{\text{DSC}}$ | $\sigma_{\text{HD95}}$ |
|---|---|---|---|---|---|---|---|---|---|---|---|---|---|---|---|
| | | | | 7.8 | 5.03 | 2.9 | 4.86 | 8.6 | 6.03 | 31.4 | 23.87 | 22.3 | 18.45 | 23.0 | 18.10 |
| | ✓ | | | 8.2 | 4.77 | 3.6 | 4.91 | 8.6 | 6.67 | 30.2 | 19.47 | 23.7 | 17.90 | 22.3 | 15.78 |
| | | | ✓ | 11.8 | 5.90 | 3.7 | 5.00 | 9.4 | 5.80 | 27.9 | 18.22 | 21.3 | 16.91 | 23.1 | 14.05 |
| | ✓ | | ✓ | 9.2 | 4.97 | 4.2 | 5.36 | 9.4 | 5.96 | 27.1 | 18.76 | 20.8 | 15.11 | 18.8 | 13.21 |
| ✓ | | | | 5.5 | 3.93 | 2.4 | 2.78 | 5.5 | 3.78 | 24.4 | 15.07 | 20.2 | 11.51 | 19.1 | 10.54 |
| ✓ | ✓ | | | 5.2 | 4.05 | 2.2 | 3.29 | 6.7 | 4.23 | 17.8 | 13.34 | 15.3 | 9.20 | 14.2 | 8.72 |
| ✓ | | ✓ | | 5.0 | 3.98 | 2.2 | 2.66 | 6.0 | 4.01 | 24.4 | 14.96 | 21.6 | 11.89 | 21.3 | 11.21 |
| ✓ | | | ✓ | 6.0 | 4.01 | 2.3 | 2.89 | 6.5 | 4.32 | 16.7 | 13.74 | 14.5 | 8.39 | 12.7 | 8.57 |
| ✓ | ✓ | | ✓ | 5.8 | 4.13 | 2.3 | 3.28 | 6.7 | 4.24 | 14.0 | 12.17 | 13.0 | 7.40 | 12.0 | 7.86 |
| ✓ | ✓ | ✓ | ✓ | 6.6 | 4.02 | 2.3 | 3.21 | 5.5 | 3.79 | 13.8 | 17.29 | 12.9 | 7.45 | 11.0 | 7.49 |

In Tab. 8 we report the structure wise mean results of the DSC and HD95 metrics for experiments where we train the model on P158 and test on the transformed test set. In Tab. 9 reports the standard deviations of the same metrics.

Table 8: DSC and HD95 with the corresponding STD across the samples, for the original test set, for training with various data augmentation strategies.

| Augmentation | | | | PZ | | | | TZ | | | |
|---|---|---|---|---|---|---|---|---|---|---|---|
| Base | MixUp | CutMix | AFA | DSC | $\sigma_{\text{DSC}}$ | HD95 | $\sigma_{\text{HD95}}$ | DSC | $\sigma_{\text{DSC}}$ | HD95 | $\sigma_{\text{HD95}}$ |
| | | | | 73.0 | 12.3 | 4.82 | 2.51 | 86.8 | 4.8 | 4.43 | 1.89 |
| | ✓ | | | 70.5 | 17.3 | 5.25 | 2.73 | 86.6 | 4.3 | 4.73 | 1.94 |
| | | | ✓ | 70.8 | 15.2 | 4.91 | 2.43 | 86.5 | 4.7 | 4.64 | 2.02 |
| | ✓ | | ✓ | 69.8 | 17.0 | 5.47 | 2.74 | 86.2 | 4.6 | 4.84 | 2.05 |
| ✓ | | | | 77.1 | 9.3 | 4.68 | 3.17 | 88.1 | 4.1 | 4.52 | 1.99 |
| ✓ | ✓ | | | 77.6 | 8.6 | 4.43 | 2.33 | 88.9 | 3.2 | 4.28 | 1.63 |
| ✓ | | ✓ | | 78.8 | 8.9 | 4.65 | 2.35 | 89.2 | 3.3 | 4.64 | 2.23 |
| ✓ | | | ✓ | 76.7 | 9.0 | 4.43 | 2.15 | 87.9 | 4.0 | 5.15 | 2.78 |
| ✓ | ✓ | | ✓ | 77.1 | 9.1 | 4.37 | 2.35 | 88.7 | 3.5 | 4.22 | 1.83 |
| ✓ | ✓ | ✓ | ✓ | 79.0 | 7.2 | 4.26 | 2.09 | 88.8 | 3.4 | 4.82 | 2.44 |

Table 9: DSC and HD95 with the corresponding STD across the samples, for the transformed test set, for training with various data augmentation strategies.

| Augmentation | | | | PZ | | | | TZ | | | |
|---|---|---|---|---|---|---|---|---|---|---|---|
| Base | MixUp | CutMix | AFA | DSC | $\sigma_{\text{DSC}}$ | HD95 | $\sigma_{\text{HD95}}$ | DSC | $\sigma_{\text{DSC}}$ | HD95 | $\sigma_{\text{HD95}}$ |
| | | | | 61.5 | 20.9 | 8.63 | 8.60 | 79.5 | 14.6 | 7.84 | 6.82 |
| | ✓ | | | 60.0 | 23.4 | 8.93 | 9.45 | 79.2 | 16.3 | 7.65 | 8.26 |
| | | | ✓ | 62.8 | 20.4 | 7.37 | 6.72 | 82.0 | 10.7 | 6.32 | 4.73 |
| | ✓ | | ✓ | 61.6 | 21.5 | 8.54 | 8.88 | 80.6 | 13.3 | 7.16 | 7.01 |
| ✓ | | | | 65.2 | 22.0 | 7.73 | 6.78 | 80.8 | 15.9 | 7.05 | 5.64 |
| ✓ | ✓ | | | 68.0 | 19.1 | 7.30 | 6.93 | 83.6 | 12.8 | 6.26 | 6.03 |
| ✓ | | ✓ | | 63.7 | 25.0 | 8.70 | 8.23 | 80.0 | 18.9 | 7.58 | 7.24 |
| ✓ | | | ✓ | 68.4 | 18.0 | 6.93 | 5.90 | 83.6 | 10.0 | 6.42 | 3.98 |
| ✓ | ✓ | | ✓ | 69.2 | 17.8 | 6.89 | 6.53 | 84.8 | 10.1 | 5.77 | 4.82 |
| ✓ | ✓ | ✓ | ✓ | 70.3 | 17.2 | 6.65 | 5.48 | 84.7 | 9.5 | 5.92 | 3.80 |

In Tab. 10 we report the structure and phase wise mean results of the DSC and HD95 metrics for experiments where we test the model trained on ACDC on M&Ms. In Tab. 11 reports the standard deviations of the same metrics.

In Tab. 12 we report the structure wise results of the DSC and HD95 metrics and the corresponding standard deviation for the experiments where we train the model on P158 on test on PX.

Table 10: DSC and HD95 performance under distribution shift for Cardiac Cine MR, when trained on ACDC and testing on M&Ms with various data augmentation strategies.

| | Augmentation | | | ES | | | | | | ED | | | | | |
| | | | | LV | | MYO | | RV | | LV | | MYO | | RV | |
| Base | MixUp | CutMix | AFA | DSC | HD95 | DSC | HD95 | DSC | HD95 | DSC | HD95 | DSC | HD95 | DSC | HD95 |
|---|---|---|---|---|---|---|---|---|---|---|---|---|---|---|---|
| ✓ | | | | 86.8 | 7.29 | 86.2 | 7.94 | 84.5 | 8.38 | 93.4 | 6.37 | 82.8 | 6.65 | 89.7 | 6.64 |
| ✓ | ✓ | | | 88.6 | 5.56 | 87.0 | 5.41 | 84.9 | 6.60 | 93.9 | 5.27 | 83.5 | 5.74 | 90.3 | 5.86 |
| ✓ | | ✓ | | 89.0 | 5.30 | 86.9 | 5.67 | 86.1 | 6.15 | 93.9 | 5.28 | 83.6 | 5.70 | 90.4 | 5.77 |
| ✓ | | | ✓ | 87.2 | 7.27 | 86.2 | 7.05 | 84.7 | 8.01 | 93.4 | 6.32 | 82.8 | 5.95 | 89.8 | 6.91 |
| ✓ | ✓ | | ✓ | 88.7 | 5.52 | 87.0 | 5.57 | 85.3 | 6.71 | 93.9 | 5.17 | 83.4 | 5.11 | 90.1 | 6.14 |
| ✓ | ✓ | ✓ | ✓ | 88.2 | 8.05 | 86.7 | 6.00 | 86.0 | 6.96 | 93.5 | 7.40 | 83.3 | 6.55 | 90.4 | 6.46 |

Table 11: The standard deviation of DSC and HD95 performance under distribution shift for Cardiac Cine MR, testing on M&Ms with various data augmentation strategies.

| | Augmentation | | | ES | | | | | | ED | | | | | |
| | | | | LV | | MYO | | RV | | LV | | MYO | | RV | |
| Base | MixUp | CutMix | AFA | $\sigma_{DSC}$ | $\sigma_{HD95}$ | $\sigma_{DSC}$ | $\sigma_{HD95}$ | $\sigma_{DSC}$ | $\sigma_{HD95}$ | $\sigma_{DSC}$ | $\sigma_{HD95}$ | $\sigma_{DSC}$ | $\sigma_{HD95}$ | $\sigma_{DSC}$ | $\sigma_{HD95}$ |
|---|---|---|---|---|---|---|---|---|---|---|---|---|---|---|---|
| ✓ | | | | 8.2 | 7.37 | 5.9 | 10.87 | 9.5 | 9.28 | 8.2 | 7.37 | 5.9 | 10.88 | 9.5 | 9.28 |
| ✓ | ✓ | | | 5.3 | 4.75 | 4.9 | 6.66 | 9.9 | 5.31 | 5.3 | 4.75 | 4.9 | 6.66 | 9.9 | 5.31 |
| ✓ | | ✓ | | 5.4 | 4.99 | 5.2 | 5.95 | 8.3 | 5.14 | 5.4 | 5.00 | 5.2 | 5.95 | 8.3 | 5.15 |
| ✓ | | | ✓ | 6.4 | 6.66 | 5.3 | 7.15 | 9.4 | 7.85 | 6.4 | 6.66 | 5.3 | 7.14 | 9.4 | 7.85 |
| ✓ | ✓ | | ✓ | 5.3 | 4.63 | 5.0 | 5.67 | 9.2 | 5.43 | 5.3 | 4.63 | 5.0 | 5.67 | 9.2 | 5.44 |
| ✓ | ✓ | ✓ | ✓ | 6.2 | 12.22 | 5.4 | 7.08 | 8.3 | 8.17 | 6.2 | 12.22 | 5.4 | 7.08 | 8.3 | 8.17 |

Table 12: DSC and HD95 with the corresponding STD across the samples, under distribution shift for Prostate bp-MRI, training on P158 and testing on PX with various data augmentation strategies.

| | Augmentation | | | PZ | | | | TZ | | | |
| Base | MixUp | CutMix | AFA | DSC | $\sigma_{DSC}$ | HD95 | $\sigma_{HD95}$ | DSC | $\sigma_{DSC}$ | HD95 | $\sigma_{HD95}$ |
|---|---|---|---|---|---|---|---|---|---|---|---|
| ✓ | | | | 55.8 | 19.1 | 10.79 | 8.57 | 85.5 | 9.7 | 5.05 | 4.33 |
| ✓ | ✓ | | | 61.7 | 14.7 | 8.61 | 5.26 | 85.6 | 9.9 | 4.59 | 1.69 |
| ✓ | | ✓ | | 66.5 | 12.8 | 7.22 | 4.00 | 87.9 | 8.2 | 4.07 | 1.80 |
| ✓ | | | ✓ | 59.1 | 16.3 | 9.94 | 7.86 | 84.5 | 10.3 | 8.40 | 12.56 |
| ✓ | ✓ | | ✓ | 61.3 | 15.7 | 8.69 | 5.41 | 85.0 | 10.0 | 6.35 | 7.64 |
| ✓ | ✓ | ✓ | ✓ | 66.4 | 12.6 | 7.43 | 4.74 | 87.8 | 6.8 | 5.05 | 5.92 |

