# OpenReview forum: "Data-Agnostic Augmentations for Unknown Variations: Out-of-Distribution Generalisation in MRI Segmentation"
_MIDL.io/2025/Conference — MIDL 2025 Poster_

### Official Review · Reviewer_4Dif · 2025-02-18

**Confidence:** 4
**Preliminary Rating:** 3
**Recommendation:** Poster
**Final Rating:** 4

**Summary:**

The paper proposes to use MixUp and Auxiliary Fourier Augmentation (AFA) as data-agnostic augmentation transforms capable of making segmentation models more resistant to unknown variations, such as bias field, ghosting, etc. The authors validate their proposal using the nnUNET framework deployed to two datasets: ACDC (Automated Cardiac Diagnosis Challenge) and P158 (Prostate 158).

If these simple data augmentation techniques actually make models more generalizable, this would have a large impact on the field of MRI segmentation.

**Strengths:**

The proposal of the paper is relatively easy to understand and follow. The authors performed numerous experiments to evaluate their proposal. These included projections of the learned features to a 2D space and analysis of the models' weight magnitudes at deeper layers in the segmentation networks.

**Weaknesses:**

My main criticism of the paper is that more details about the experiments, particularly the hyperparameters used, are not there. In essence, I find it would be hard to replicate the work without a better description of the methods.

**Detailed Comments:**

- How do you determine the corruption severity level?
- Standard deviations should be reported in the results tables. Using three colours (red, blue, black) and bold to distinguish the results makes the table harder to read.
- Why use a paired t-test? Is the test performed using average results across each fold or individual metrics for each sample used during testing?

**Justification Of The Final Rating:**

My main concern was reproducibility issues. The authors made their code publicly available and provided mostly satisfactory answers to my concerns.

The authors just need to clarify how the k-space pertubation happens. The code does not match the paper equations.

**Justification Of The Preliminary Rating:**

Though this is an interesting work with potential implications for MRI segmentation, I am afraid that more details are needed for the scientific community to be able to reproduce and build on top of the work.

**Questions To Address In The Rebuttal:**

- More details about the hyperparameters and how the proposed data augmentations are used to improve the robustness of the segmentation network.
- Variable alpha in the AFA equation. Is this a real or complex-valued number? Please clarify.
- Could you show sample images after the two proposed data augmentations (MixUp and AFA)?

---

> ### Author Response · Authors · 2025-03-08
>
> We thank you for taking time to review our work, and your questions. Below, we provide a point-by-point response to the questions asked by the reviewer.
>
> ## Determination of the Severity
>
> Following the approach of ROOD-MRI [1], we defined five distinct severity levels for each transformation. We have at an early stage consulted with multiple MR technicians with experience on multiple sites of MRI, who indicated that these images closely resembled what would happen in clinical practice. This process and choice has been clear in the revised manuscript, Section 2.2.
>
> ## Standard Deviations and Tables
> We thank the reviewer for pointing out the need for standard deviations in Tables 1 and 2. We include the standard deviations in the revised manuscript under Appendix (E) along with structure wise results. We will improve the readability of the tables and the colours used such that the results are obvious at a glance.
>
> ## Use of paired t-test.
> We appreciate the opportunity to clarify our statistical methodology and explain why this approach is well-suited for
> evaluating the performance of our models.
>
> The default running operation of nnUNet is to perform a 5-fold cross-validation and then use the ensemble of the five
> models to make predictions on the test sets. Therefore, in our experiments, the paired t-test is performed using individual metrics for each sample during testing, rather than averaging results across folds. This approach also offers a better statistical test when we deal with patient scans with high variability, like samples in our transformed test set. Comparing cross-validation averages, might obfuscate this granularity.
>
> ## Hyperparameters
> A comprehensive list of hyperparameters used in our experiments, for our augmentations and training, is provided in the
> revised manuscript under Appendix (C). The program code for our experiments is also now made publicly available to
> ensure the reproducibility and the ease of adoption of our methods: https://github.com/MIAGroupUT/augmentations-for-the-unknown.
>
> ## \alpha in AFA augmentation.
> This is a real value that controls the magnitude of the augmentation. This is sampled from an exponential distribution as described in the original AFA paper. We have made clarification on this in revised manuscript in Sec. 2.3.
>
> ## Examples of MixUp and AFA
> We have included examples of MixUp and AFA augmentations in the revised manuscript under Appendix (B).
>
> - Misc
>   - We have strengthened our discussion to motivate further how these augmentations are used to improve the robustness of the segmentation network.

---

> > ### Comment · Reviewer_4Dif · 2025-03-10
> > **Still about alpha...**
> >
> > Thank you for the detailed answers.
> >
> > The description of alpha is still not 100% clear to me. If you add a real value (i.e., alpha) to a random position in k-space. Then, k-space is no longer symmetric, and the inverse Fourier Transform (iFT) of said k-space is complex-valued instead of real-valued.
> >
> > Looking at the code the authors provided, it seems they are extracting the real portion of the iFT. What is the justification for this? Also, it doesn't match the description in the paper, which should be fixed if this is the case.

---

> > ### Author Response · Authors · 2025-03-12
> >
> > We thank the reviewer for their time and their in depth comments on our work. We further discuss the points raised.
> >
> > ## Real Fourier Transform and Symmetry in k-Space
> >
> > You are correct that we only consider the real component of the Fourier Transform, but this is because by using only the real inverse fourier transform which takes only the half of the frequency spectrum, therefore, the k-space has not been made asymmetric. We will further clarify the use of real Fourier transform in the text.

---

### Official Review · Reviewer_HFY4 · 2025-02-20

**Confidence:** 5
**Preliminary Rating:** 1
**Final Rating:** 2

**Summary:**

The paper explores the impact of data agnostic augmentations in medical image segmentation for handling out-of-distribution (OOD) generalization in MRI segmentation tasks. Traditional augmentation strategies may fail to address complex real-world variations, and this paper use the existing MixUp and Auxiliary Fourier Augmentation (AFA) to improve robustness against unknown variations. The paper is conducted on cardiac and prostate datasets

**Strengths:**

-Comprehensive evaluation: This paper accesses its effectiveness in nnU-Net pipeline for cardiac cine MRI and prostate MRI segmentation tasks.

-This paper validates 14 different image corruptions and evaluates model performance on datasets from different domain/imaging site. For example, use ACDC and P158 as training and test on M&Ms and ProstateX, respectively

**Weaknesses:**

Related to Table 1:
While I understand the meaning of "transformed," many of these image corruptions, such as bias field distortions could be corrected with existing image preprocessing techniques. However, some of them are hard to be corrected during post-processing, and these should be discussed in detail.

For other corruptions, such as Rician noise, spike noise, and k-space subsampling artifacts, I acknowledge that they exist in MRI. However, do they normally appear in real-world clinical practice? What's the severity? These artifacts are not commonly reported in recent studies because modern MRI acquisition techniques have significantly improved. While testing robustness against synthetic distortions is useful, I question the real-world applicability of training segmentation models on heavily corrupted data, especially when such extreme distortions are rarely encountered in clinical workflows.

Besides the transformed, the other results are comparable.


Related to Table 2:
Another concern is based on the real OOD tests in table 2, where the augmentations are validated with domain gaps, particularly in prostate segmentation. However, due to the inherent MR Imaging properties, especially there is no content to show the domain gaps between these datasets, this Table is hard to be evaluated. Beside this, the prostate segmentation has its own challenging where high anatomical variabilities are presented.


Additionally, the authors mention Apparent Diffusion Coefficient (ADC) maps, but it is unclear whether these were used as inputs. If ADC maps were not included, it would be useful to explain the reasoning behind this decision, as ADC is often critical for prostate MRI analysis. If they were used, specifying the b-values would be important, as different b-values significantly affect image contrast and segmentation performance

Overall:

Limited novelty. This work is build on the existing augmentation methods

No compared augmentation methods designed for medical images.

As author discussed, the PRIME is not applied. Additionally, Manifold MixUp, CutMix, and etc..

Based on the above settings, more datasets should be used to test robustness, as the main and only contribution should focus on additional experiments. The current limitation restricts applicability to other modalities (e.g., brain, liver, lung MRI)

Importantly, while this may seem trivial, it is still important. The authors should include a short introduction on the original nnU-Net settings for reference, such as whether 2D, 3D, cascaded, or ensemble configurations were used.

Computational cost and convergence speed are not addressed. even this is a minor point, it is important for nnU-Net, which performs well but requires relatively longer training times."

While MixUp and AFA are shown to improve robustness, there is no in-depth justification for why they outperform standard augmentations in MRI. Additionally, there are no structure-wise results, which are crucial in medical imaging

Figure 3 may not reflect real-world generalization, why not to perform it on real datasets?

**Detailed Comments:**

please see weaknesses

**Justification Of The Final Rating:**

This work has undergone extensive validation, but my rating is based on the motivation and implementation details.

The implementation details are important for this paper as they could impact the overall findings. The author has listed the detailed information as a reference, and the results for domain shift are conducted on real MR images.

However, I don’t like that the main work relies on synthetic results instead of using real images without proper validation in the medical imaging domain.

**Justification Of The Preliminary Rating:**

This paper uses a simple yet effective augmentation strategy for different medical image segmentation tasks. However, my rating is primarily based on its clinical impact, results analysis, experimental setup, and the number of datasets used. There are no impressive valuable findings in the paper of their results. They should either improve their analysis approach or include more datasets to strengthen the evaluation. Since there is no option of reject, I consider this paper to be below my threshold for a weak reject.

**Questions To Address In The Rebuttal:**

please see weaknesses

---

> ### Author Response · Authors · 2025-03-08
>
> We thank the reviewer for their thoughtful comments. Below, we provide a point-by-point response to the main issues raised by the reviewer.
>
> ## Real-world Applicability
> The reviewer asks, to what extent the distortions that we apply in the transformed data set reflect what can be expected in clinical practice.
>
> First, we would like to remark that we are not *training* on these distortions, but only using them to generate a challenging test set. In this test set, we vary the severity of these distortions between 1 and 5. We have at an early stage consulted with an MR technician, who indicated that (1) these images closely resembled what would happen in clinical practice, and (2) for most distortions, a severity level of 3 or higher would require rescanning.
>
> Hence, we agree with the reviewer that distortions with such high severity levels are unlikely in the diagnostic images that the radiologist might see. However, we do include for two reasons. First, they allow us to quantify model robustness better. Second, by developing algorithms robust to these higher severity levels, the number of images that need to be retaken might be reduced, thereby enhancing the efficiency of MR scanning.
>
> As the reviewer writes, some of the distortions that we apply can indeed be corrected in pre-processing (e.g. bias fields), while others cannot. In the paper, we now focus more on the latter category.
>
> ## Comparable Results on Non-Transformed Test Set
> We thank the reviewer for noting that our method achieves comparable performance on the original dataset while
> significantly improving robustness to corrupted data. This is a critical strength of our approach, as many methods that
> aim to improve robustness to corrupted data often trade off performance on the original dataset. Our method maintains performance on the original dataset while improving robustness to corrupted data, unlike methods that trade off clean-data performance for robustness. We highlight this strength in Sec.  ​4. For example:
>
> - Methods that rely heavily on aggressive data augmentation or noise injection can sometimes overfit to the augmented
>   data, leading to suboptimal performance on clean data.
> - Similarly, techniques designed to handle specific types of corruption (e.g., noise or artifacts) may fail to
>   generalize to the original data distribution, resulting in a loss of efficacy.
>
> Our approach, on the other hand, is designed to improve robustness to a wide range of transformations while maintaining
> high performance on the original dataset, all while not mimicking the distribution of the transformed data.
>
> ## Latent Space Plot for real world distortions
> To the best of our knowledge, there are no real-world distorted datasets with annotations of the masks. Therefore, our next best alternative is to use generated variations, which are realistic and controlled, to evaluate robustness.
>
> ## Extra Experiments
> In response to the reviewer’s suggestion, we have conducted additional experiments with CutMix and one additional dataset for Brain MRI using the ADNI dataset and an additional OOD distribution Hippocampus segmentation dataset HFH using 3d fullres nnU-Net. This also showcases that these augmentation strategies are not limited to be applied to other modalities. Due to time constraints, these are in Appendix I and will be included in the camera-ready version.
>
> ## More Augmentation Strategies
> We clarify why methods like PRIME and Manifold MixUp were not included and discuss their relevance for future work in Sec. 4.
>
> ## Run Configuration, Computational Cost, and Code Availability
> Hyperparameters and run configurations are now provided in Appendix C.
>
> ## Value of Results
> Our proposed augmentation strategies- MixUp and Auxiliary Fourier Augmentation- are designed to enhance robustness without compromising performance on the original dataset. This is achieved by promoting feature compactness and separability, which improves generalization across both clean and corrupted data. Our results demonstrate that this approach strikes a balance between robustness and accuracy, making it highly suitable for real-world clinical applications where models must perform well across a wide range of conditions. We believe that this finding is important to the community, which frequently phases such scenarios in real-world applications and makes heavy use of the nnU-Net model, which, as we show, can provide consistently better results using these simple and cheap augmentation strategies.
>
> ## Misc
> - Limited Novelty: We acknowledge no novel augmentation technique, but we submit to the Validation and Application track for analyzing augmentations under distribution shifts.
> - Medical-specific augmentations: nnU-Net covers current best practices for MR augmentations; we show further improvement with our data-agnostic augmentations.
> - Structure-wise results are added to Appendix E.
> - ADC maps: They are used for prostate gland segmentation, with b-values added to Sec. 2.1.

---

> ### Comment · Reviewer_HFY4 · 2025-03-11
>
> I thank the authors for their responses in addressing my comments.
>
> However, I have two key points I would like to discuss:
>
> 1. Real-world application
> My concern refers to the test set rather than the training process. Additionally, post-processing refers to steps performed at the scanner level, whereas the artifacts introduced in this study could be also addressed through pre-processing at the image level. If standard image processing techniques can already correct these artifacts, then it is unclear why training a model to handle them is necessary.
>
> The authors acknowledge that radiologists may not even encounter some of these high-severity distortions. Based on my experience, such artifacts are rarely observed in real MR images. This also ties into the "technician" reference made by the authors, as MRI technicians could correct these issues, preventing them from affecting diagnostic imaging.
>
> The motivation for this study is still not fully on my side. If the focus were on a specific patient group (e.g., patients who cannot remain still in the scanner for long periods, leading to compromised image quality; or requiring rescans, as the authors indicated) or on another imaging modality such as ultrasound or some potable imaging device, where image quality varies significantly, the problem setup would be clearer. However, the constraints of k-space acquisition in MRI limit such variability.
>
> This paper builds upon previously proposed AFA augmentation methods, which align with k-space. However, its motivation does not seem well-suited for some medical applications, as it appears to have been directly adapted with minimal modifications. Nevertheless, the current validation, along with the proposed revisions, helps address some of these limitations.
>
> 2. I think that 200 epochs are sufficient for training. The models may be undertrained. nnU-Net is widely used and recognized for its superior and robust performance in radiology, particularly in medical imaging challenges and competitions. However, I am not aware the top-performing methods use such a short epoch number. This raises concerns about whether the models in this study have reached their full learning capacity, which in turn affects the validity of the conclusions drawn.
>
> Since there is no page limit for appendix, I recommend adding dataset information in the captions of some tables to improve clarity.
>
> Missing modality considerations would be a valuable future direction for MRI research.
>
> The description of b-values is not accurate, please double-check and revise it. Also, the usage of bpMRI might either be incorrect or too vague. I suggest referring directly to the dataset name for clarity.
>
> I appreciate the authors' efforts in strengthening this work without significantly changing the main structure and content. The proposed revisions will provide greater clarity and insights for readers.

---

> > ### Author Response · Authors · 2025-03-12
> >
> > We thank the reviewer for taking the time to go through the revision and the points of discussion.
> >
> > ## 1. Real-world Application
> > First, regarding post-processing (on the MRI scanner) and pre-processing (in the image domain), we would like to clarify that standard image processing cannot correct most artifacts, as discussed in the introduction of the paper. Bias fields can potentially be corrected by region-based normalisation, but this is also not standard. Many artifact corrections and de-noising methods are highly ill-posed, and therefore can not be corrected by simple image processing. In the appendix, we included results for a wide range of artifacts that cannot be fixed with image processing techniques. For instance, random motion which cannot be fixed by simple image processing techniques or by rescanning (because of heart arrhythmia motion of lungs, etc.), further highlighting the need for robust models that can handle such variations.
> >
> > Second, we would like to clarify that we have aimed to develop a general approach to improved robustness in MRI segmentation and consider a wide range of potential artifacts, some of which might not be very common. By considering cardiac cine MRI, prostate MRI, and (currently in the appendix) brain MRI, we demonstrate the general nature of our findings. However, we agree with the reviewer that not all of these artifacts would be relevant or realistic for specific subpopulations or applications. However, we believe that the findings of this paper are also relevant for such specific applications and can be used to improve segmentation robustness. For example, in patient groups where rescans are expensive or impossible, a more robust model could reduce the need for rescans.
> >
> > Regardless of the specific application, there are no downsides to having a more robust model. As AI models are deployed in increasingly diverse and unpredictable real-world settings (unknown variations), robustness becomes a critical factor in ensuring reliable performance across varying conditions.
> >
> > ## 2. Training Duration
> >
> > We acknowledge the reviewer’s concern regarding the training duration of 200 epochs and its potential impact on model performance. Here, we provide further clarification:
> >
> > -  Training models for longer can lead to overfitting to the training distribution, which may negatively impact out-of-distribution generalization. Our empirical results demonstrate that the models achieve competitive or superior performance compared to top submissions in challenges. For instance, ACDC, where the best performing model made available by the original authors of nnU-Net [1] comprised of a 2D and 3D full-resolution U-Net reaches a performance of DSC=0.9228 as opposed to DSC=0.9250 on the model we train with only base nnU-Net augmentations. This performance is also comparable to other state-of-the-art methods since then, as discussed in the study [2].
> >
> > - Other studies have also used a smaller number of epochs of 300 [3], 150 [4], 500 and 200 [5] and an optimal number of epochs for nnU-Net could warrant its own validation study. While longer training might yield marginal improvements, it could also increase the risk of overfitting, especially in scenarios with limited data. We add plots here of the loss and evaluation metric (average DSC) automatically generated by nnU-Net to show the training dynamic of the nnU-Net with only base augmentations and with MixUp stagnates in terms of validation loss and evaluation metrics for both ACDC P158 well before 200 epochs. We believe that the current training duration, supported empirically in our results, results in no loss of learning.
> >
> > Loss and Evaluation Metric Curves for nnU-Net with only base augmentations for ACDC: https://imgur.com/a/kcGD1Bv
> >
> > Loss and Evaluation Metric Curves for nnU-Net with base augmentations and MixUp for ACDC: https://imgur.com/a/NH8BzbX
> >
> > Loss and Evaluation Metric Curves for nnU-Net with only base augmentations for P158: https://imgur.com/a/FXholRH
> >
> > Loss and Evaluation Metric Curves for nnU-Net with base augmentations and MixUp for P158: https://imgur.com/a/av0CX8G
> >
> > [1] Isensee, F., et. al (2021). Pretrained models for 3D semantic image segmentation with nnU-Net (2.1). doi:10.5281/zenodo.4485926
> >
> > [2] Isensee, F., et. al (2024). nnU-Net Revisited: A Call for Rigorous Validation in 3D Medical Image Segmentation. doi:2404.09556.
> >
> > [3] Raab, F.,et. al (2023). Investigation of an efficient multi-modal convolutional neural network for multiple sclerosis lesion detection. doi: 10.1038/s41598-023-48578-4
> >
> > [4] Heinrich, M. P., & Hagenah, J. (2023). Make nnUNets Small Again.
> >
> > [5] Xue, X., et. al (2024). A deep learning-based 3D Prompt-nnUnet model for automatic segmentation in brachytherapy of postoperative endometrial carcinoma. doi: 10.1002/acm2.14371

---

> > > ### Comment · Reviewer_HFY4 · 2025-03-12
> > >
> > > 1. I just went through Figure 4 and still think more than half of them don't make sense. By the way, what do you mean by "simple" image processing
> > >
> > > 2. Your initial learning rate is 0.01 for 200 epochs, which raises my concern about it being too large.
> > >
> > > 3. That sounds good.

---

> > > > ### Comment · Reviewer_U6tW · 2025-03-12
> > > >
> > > > To pitch-in here (even though it seems my esteemed colleague already has provided a final rating), what I liked about the paper is the systematic analysis, showing that seemingly nonsensical yet simply data augmentation techniques (e.g., MixUp), have such a consistent effect on a broad class of (realistic or not) perturbations; and corroborated by the measurable differences in learned features (Fig 12, 15 notably).
> > > >
> > > > So to me, even if a lot of those synthetic noise are unrealistic, the fact that the networks trained with MixUp/AFA handle it much better "out of the box" is a very interesting insight, that I was not expecting; which is enough in my opinion to deserve a spot at the conference. A journal version should be much more thorough and then follow Reviewer HFY4 recommandations, notably testing on real data (and ideally different modalities/tasks).
> > > >
> > > > If you had asked me about the value of MixUp before reading that paper, my answer probably would have been along the line of "bweh", while now I realize there is some deeper value and mechanisms at play that deserve my attention, all while being (potentially) to be an "easy" improvements for generalization on some datasets and tasks in the field. [Notably Ultrasounds where I'd be curious to see if we observe the same effects.]

---

> > > > > ### Comment · Reviewer_HFY4 · 2025-03-12
> > > > >
> > > > > just different flavors, but the details are definitely required.

---

> > > > ### Author Response · Authors · 2025-03-13
> > > >
> > > > We thank the reviewer for their continued engagement, for investing time in our work, and for recognizing the thoroughness of the validation in the updated rating.
> > > >
> > > > To address the previous comments:
> > > >
> > > > ## Transformations
> > > > To provide further clarity, we will add a description of each transformation in the appendix. This will allow researchers to better understand the rationale behind each transformation and enable them to pick and choose variations that are most relevant for their specific applications. We closely followed the ROOD-MRI framework and methodology for our transformations, which has been validated in the medical field and which we cite in Section 2.2 [1].
> > > >
> > > > ## Learning Rate
> > > >
> > > > - We believe the learning rate of 0.01 used in our study is not too large for 200 epochs, as other nnU-Net models in literature are even trained with learning rates as high as 0.1 for similar epochs [2].
> > > > - We also use the default nnU-Net learning rate scheduler (polynomial decay with power=0.9), as mentioned in the run configurations. This scheduler gradually reduces the learning rate over time, ensuring stable convergence during optimization.
> > > > - Furthermore, the current training setup converges and performs similarly or is an improvement to other configurations.
> > > >
> > > >
> > > > [1] Boone, L., et. al (2023). ROOD-MRI: Benchmarking the robustness of deep learning segmentation models to out-of-distribution and corrupted data in MRI. Neuroimage, 278, 120289. doi: 10.1016/j.neuroimage.2023.120289
> > > >
> > > > [2] Xue, X., et. al (2024). A deep learning-based 3D Prompt-nnUnet model for automatic segmentation in brachytherapy of postoperative endometrial carcinoma. doi: 10.1002/acm2.14371
> > > >
> > > > ------
> > > > Once again, we appreciate the reviewer’s time, engagement and constructive feedback, which has helped us improve the clarity and rigour of our work.

---

> > ### Author Response · Authors · 2025-03-12
> > **Misc Points**
> >
> > ### Additional Points:
> >
> > -   Dataset Information in Captions: We will include additional dataset details in the captions of relevant tables in the appendix to improve clarity in the revised manuscript.
> >
> > -   Missing Modality Considerations: We agree that this is a valuable direction for future research and will highlight this as a future work in the revised manuscript.
> >
> > -   b-values and bpMRI Description: Below are the exact statements in the original dataset. We will correct and revise the description of b-values and clarify the usage of bpMRI, referring directly to the dataset name for precision.
> > 	- P158: These were using the DWI image acquired at the same b-values. > "The software pre-installed on the MRI scanner (version VE11A) was used to calculate an ADC map from b-values ranging from 50 to 1000 s/mm³ and a high b-value of 1400 s/mm³."
> > 	- PX: > "DWI series was acquired, ..., 3 b-values (50, 400, and 800 s∕mm2), and a computed apparent diffusion coefficient map."
> >
> >
> > We appreciate the reviewer’s constructive feedback and their acknowledgment of our efforts to strengthen the work.

---

### Official Review · Reviewer_U6tW · 2025-02-21

**Confidence:** 5
**Preliminary Rating:** 4
**Recommendation:** Oral, Poster
**Final Rating:** 5

**Summary:**

This paper benchmarks data-agnostic data augmentation (i.e., the augmented images are not necessarily realistic) to test if and how much they can improve generalization performances of trained segmentation networks.

Overall the paper is thorough, clear, and nicely presented; interesting insights are demonstrated through their experiments.

**Strengths:**

- The paper topic is timely and well motivated, and is a great fit for the conference
- Overall the paper was pleasant to read
- Some of the conclusions and figures were thoughts provoking (notably Fig. 11 and 12 which I really liked).

**Weaknesses:**

- Some clarifications are required or would be welcome:
    - notably adding STD to the metrics reported in Table 1 and 2 would be very useful;
    - as well as clarifying the number of runs (how many times the networks were trained?)

**Detailed Comments:**

Misc:
- for reproducibility, it would be good to explicit which implementation of the HD95 was used [1]
- similarly I would release the code of the paper, that would strengthen the paper and its impact
- the figures placements with respect to their sections (notably in the appendix) should be improved
- at the moment I cannot read Figure 10, I do not know how to interpret it (notably because the misclassification color (yellow) is used in the ground truth column)
- Fig. 10 still: crop the images around the object for easier visualization for the reader

[1] https://arxiv.org/pdf/2410.02630

**Justification Of The Final Rating:**

I am happy with the rebuttal and I think that this paper clearly fits the scope of MIDL and its aims.

That said, I strongly encourage authors to polish their final manuscript, especially when it comes to reporting results. It can only increase the visibility and impact of their work, and the value to the MIDL community as a whole.

**Justification Of The Preliminary Rating:**

My first rating is quite conservative but I expect to raise it after rebuttal, provided the authors engage in it. Note that some of my questions for the rebuttal are geared toward a journal extension, and I am not necessarily expecting it to make it to the conference paper (notably for the questions about the optimizer).

**Questions To Address In The Rebuttal:**

### Label interpolation
> We adapt the original MixUp Setup, in which $y_i$ is a one-hot encoded sample label, to segmentation by linearly interpolating one-hot encoded segmentation masks

This is not clear to me, can you clarify what is done exactly? Also, notice that both the Cross-Entropy loss and Dice loss _can_ handle probabilities as input for the label, so I am not sure why the adaptation is needed. I would even hypothesize that this adaptation can have negative effects with respect to the network calibration.

### Reproducibilty and variability under random initialization
Notably when looking at Figure 3 (very interesting), I am wondering how much it varies across different runs/random initializations. Providing in the appendix the same figure across e.g., 5 runs, would be useful for the reader to get a better idea

### Impact of the optimizer
It is my understanding that the authors used nnU-Net for their experiments (this is good), which would mean that standard SGD was used as optimizer. Looking at Figures 11 and 12, I am wondering what would happen if let's say, Adam, was used as optimizer: would the implicit and explicit normalization of the layers weights reduce or even suppress the observed effect?

More broadly, would the authors think that their conclusions would hold under other optimizers?

**Special Issue:**

Yes

---

> ### Author Response · Authors · 2025-03-08
>
> Thank you for your thoughtful review, interesting questions and for recognizing the importance of our work.
>
> ## Label interpolation
> In the original MixUp formulation for classification tasks, images and one-hot encoded labels are both linearly interpolated. Here, we use the exact same strategy, but instead of linearly interpolating between two labels, we interpolate between two one-hot encoded segmentation masks. To be precise, we interpolate as follows:
>
> - Two input images $X_1$ and $X_2$ are linearly interpolated as $X_{mix}=\lambda X_1+(1−\lambda) X_2$, where
>   $\lambda$ is sampled from a Beta distribution.
> - Two segmentation masks $Y_1$ and $Y_2$ are linearly interpolated
>   as $Y_{mix}=\lambda Y_1+(1−\lambda) Y_2$ using the same $\lambda$.
>
> As the reviewer states, Cross-Entropy and Dice loss can use probabilities as inputs, and this is indeed what we provide here. We have further clarified this in the paper.
>
> ## Reproducibility and Variability Under Random Initialization
> We have generated the latent space representation for four other runs (hence 5 in total), one for each fold that we train in our five-fold cross-validation
> training process. We observed similar results, and we have included this in the revised manuscript in the Appendix F.
>
> ## Impact of the Optimizer
> We thank the reviewer for this insightful question.
>
> We believe that our conclusions regarding the effectiveness of MixUp and Auxiliary Fourier Augmentation hold broadly
> across different optimizers. While the specific magnitude of regularization effects may vary, the underlying
> mechanisms—promoting feature compactness and separability—are optimizer-agnostic. These mechanisms are driven by the
> augmentations themselves rather than the optimizer's normalization properties. Therefore, we expect that the benefits of
> MixUp and Auxiliary Fourier Augmentation would be preserved across different optimizers.
>
> Additionally, Adam as an optimizer has been shown to be less sensitive to hyperparameters and initialisation of the
> network than SGD as it is an adaptive method. However, it is not necessarily better in terms of the generalization property
> of the model [1].
> In view of time, we have not been able to perform these experiments yet but consider them an interesting addition to the camera-ready version.
>
> [1] Zhou, P., Feng, J., Ma, C., Xiong, C., Hoi, S. C. H., & E, W. (2020). Towards Theoretically Understanding Why SGD
> Generalizes Better Than Adam in Deep Learning. Proceedings of the 33rd Advances in Neural Information Processing
> Systems.
>
> ## Misc
>
> - Standard Deviation
>     - Standard deviation and structure wise results are added to the Appendix (E).
> - Number of Folds:
>     - We train our models using a 5-fold cross-validation strategy. Note that this is mentioned in the paper (Sec. 2.4).
> - Implementation of HD95:
>     - We use the implementation of the Hausdorff Distance with a 95th percentile (HD95) from MONAI [2]. This choice has been made clear in the paper.
> - Hyperparameters and Code Availability:
>     - A comprehensive list of hyperparameters used in our experiments, for our augmentations and training, is provided in the revised manuscript under Appendix (C). The program code for our experiments is also now made publicly available to ensure the reproducibility and the ease of adoption of our methods: https://github.com/MIAGroupUT/augmentations-for-the-unknown.
> - Figure Placements and Improvements
>     - We improve the placement of the figures in the revised manuscript for the best readability.
>     - Figure 10 will be re-made with the feedback suggested by the reviewer, so that there are no confusions.
>
> [2] Cardoso, M. J., Li, W., Brown, R., Ma, N., Kerfoot, E., Wang, Y., ...Feng, A. (2022). MONAI: An open-source
> framework for deep learning in healthcare.

---

> ### Comment · Reviewer_U6tW · 2025-03-12
>
> I thank the authors for their reply, and I am happy that multiple runs were performed, producing notably Figure 12. (BTW typo: "the one wirtten" in its caption.) Given how much randomness there is in deep learning (random sampling, random augmentation, random initialization, at least), I find that it is a very strong result to see the same patterns on the feature representations appear consistently across runs. I would make that part prominent within the paper.
>
> Concerning the standard deviation, added as separate Tables,  in the Appendix, I find this choice curious, to say the least. Looking at Table 1 there is clearly the space to add the std on the size. Notice that 0.XXX when reporting the DSC is redundant: XX.X is as clear. So Table 1 could contains numbers such as $89.1{\small \sigma 7.8}$. Right now it is very clumsy to scroll through, 20 pages apart, just to get this info, which in my opinion is non-optional when reading results.
>
> Concerning the clarification for the interpolation, I have no further question.

---

> > ### Author Response · Authors · 2025-03-12
> >
> > We sincerely thank the reviewer for their time, and their thoughtful questions and constructive feedback, which have been invaluable in strengthening our work.
> >
> > Below, we outline the changes we will make for the final manuscript:
> >
> > ## Feature Representations
> > We thank the reviewer for suggesting the additional runs to be included in the paper, and we are glad that you also find the consistency of feature representations across multiple runs to be a key finding. We will make this part more prominent in the paper in our discussion.
> >
> > ## Presentation of Standard Deviations
> > We appreciate your suggestion to improve the presentation of standard deviations. We agree that including the standard deviation directly in the Table 1 and 2, as you proposed "XX.X $\sigma$ Y.Y"), is more reader-friendly and avoids the need to scroll through the appendix for this information. We will revise Table 1 and 2 accordingly to enhance readability and ensure that these results are presented in a clear and concise manner.
> >
> >
> > ### Misc
> > We are glad that the clarification regarding interpolation addressed your concerns.
> >
> > Once again, we would like to express our gratitude for your time, constructive feedback and efforts through the review process.

---

### Author Response · Authors · 2025-03-08
**General Response to Everyone**

We thank all the reviewers for their valuable and thorough feedback. We thank the reviewers for their remarks and for recognizing that our work is easy-to-follow, timely motivated, pleasant to read and understand and a great fit for the conference. It is very encouraging that the reviewers found our work supported by comprehensive evaluation with extensive evaluation of performance on from different domain and imaging sites, paired with thought-provoking analysis, which show effectiveness to improve OOD generalization and robustness.

We appreciate the reviewers' constructive feedback and suggestions. We address the concerns raised by the reviewers in the revised manuscript. We respond to the various points and reviewers' feedback in separate comments.

---

### Author Rebuttal · Authors · 2025-03-08

**Rebuttal:**

We thank the reviewers for their thoughtful feedback and constructive comments. We have revised the manuscript to address all major concerns, including clarifications on some parts of our methodology, improved the discussion and added preliminary additional experiments. All changes in the manuscript are highlighted by the use of teal colour.

**Supporting Material:**

/attachment/40090a8b4cf10b1370f6b45d9fe6c857fd510d16.pdf

---

### Meta-Review · Area_Chair_rkQy · 2025-03-21

**Recommendation:** Accept (Poster)
**Confidence:** 5

**Metareview:**

This paper investigates using data-agnostic augmentations (MixUp and AFA) to improve robustness in MRI segmentation. The paper is clear, well-organized, and presents comprehensive experiments, including robustness evaluations under multiple corruption types (like ImageNet-C) and domain shifts. The analysis of domain shift is meaningful and relevant to the field.

Strengths:
- Comprehensive experiments with clear results.
- Relevant topic with practical significance for medical image segmentation.

Weaknesses:
- Limited novelty and insights.
- Limited comparisons with other augmentation methods.

Additionally, I have a separate comment. It is not entirely clear why these two augmentations (MixUp and AFA) were specifically chosen. Do they represent certain types of augmentation strategies? What guidance do they provide for selecting other augmentation methods? Currently, many of the conclusions are relatively isolated and specific to these two augmentations. While I understand that the paper has conducted extensive experiments, to further increase its impact, it would be beneficial to benchmark more augmentation methods and analyze the underlying factors behind their performance. This would enable a deeper and more general understanding of domain shifts.